# A Single Goal is All You Need:
## Skills and Exploration Emerge from Contrastive RL without Rewards, Demonstrations, or Subgoals

**Grace Liu**[*]     **Michael Tang**     **Benjamin Eysenbach**
Princeton University

## Abstract

In this paper, we present empirical evidence of skills and directed exploration emerging from a simple RL algorithm long before any successful trials are observed. For example, in a manipulation task, the agent is given a single observation of the goal state (see Fig. 1) and learns skills, first for moving its end-effector, then for pushing the block, and finally for picking up and placing the block. These skills emerge before the agent has ever successfully placed the block at the goal location and without the aid of any reward functions, demonstrations, or manually-specified distance metrics. Once the agent has learned to reach the goal state reliably, exploration is reduced. Implementing our method involves a simple modification of prior work and does not require density estimates, ensembles, or any additional hyperparameters. Intuitively, the proposed method seems like it should be terrible at exploration, and we lack a clear theoretical understanding of why it works so effectively, though our experiments provide some hints.

Videos and code: `https://graliuce.github.io/sgcrl/`

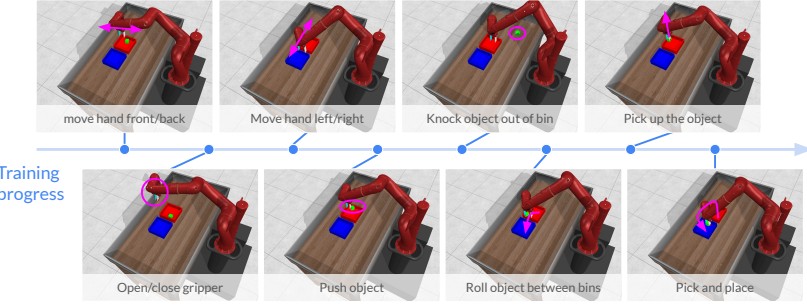

Figure 1: **Skills and Directed Exploration Emerge.** In this task, we provide the agent with a single goal observation where the green block is in the left bin. The agent never receives any rewards (not even sparse rewards). Throughout training, the agent learns skills that increase in complexity. Easier skills seem to enable the agent to unlock more complex skills: moving the hand is a prerequisite for pushing the object; closing the gripper is a prerequisite for picking up the object, which is a prerequisite for moving the object to the left bin.

## 1 Introduction

Exploration is one of the grand challenges in reinforcement learning (RL) (Thrun, 1992). Effective exploration algorithms would enable RL agents to solve long-horizon, sparse reward problems with minimal human supervision: no need for dense reward functions, demonstrations, or hierarchical RL. While there is a long history of exploration methods, even today's best methods fail to explore in settings with sufficiently sparse rewards, and the complexity of sophisticated exploration techniques means that most researchers today employ limited exploration methods (e.g., adding random noise to actions (Heess et al., 2015; Lillicrap et al., 2015; Mnih et al., 2013; Fujimoto et al., 2018)).

In this paper, we focus on a specific type of RL problem (Kaelbling, 1993; Chane-Sane et al., 2021; Liu et al., 2022): the agent is given an observation of the single desired goal state, which

---

[*]Correspondence to: Grace Liu <gliu2@andrew.cmu.edu>

it tries to reach. This problem setting captures many practical problems, from cell biology (grow a certain cell type) to chemical engineering (create a specific molecule) to video games (navigate to the final room). However, this problem setting is exceedingly challenging for standard RL methods, as the agent does not receive any reward feedback about *how* it should solve the task. In continuous settings, the agent will never reach the goal *exactly*, so no reward signal is ever observed. Because of the difficulty of this exploration problem, prior work typically assumes that a human user can provide a dense reward function (Yu et al., 2020; Hansen-Estruch et al., 2022) (or distance metric/threshold (Venkattaramanujam et al., 2019; Plappert et al., 2018a; Chane-Sane et al., 2021)) or a set of easier training goals[1] (Eysenbach et al., 2022). However, constructing these reward functions or easier goals is challenging (Liu et al., 2022; Hadfield-Menell et al., 2017) and stymies potential applications of RL: a chemist who wants to synthesize one particular molecule would have to write down several "easier" molecules for the RL agent to reach. Our paper lifts the assumptions of prior work by considering a setting that is easier for human users but significantly more challenging for RL agents: a single goal state is provided and is used for both training and evaluation.

We present a simple RL algorithm where skills and directed exploration emerge long before any successful trials are ever observed. For example, in a manipulation task where the agent is given a single observation of the goal state (see Fig. 1), the agent ends up learning skills for moving its end-effector, then for pushing the block, then for lifting the block, and finally for picking up the block. These skills are learned before the agent has ever succeeded at placing the block in the correct location and without any reward functions, demonstrations, or manually-specified distance metrics. Once the agent has learned to reliably reach the goal state, it slows exploration. Implementing this method involves a simple modification of prior work and does not require density estimates, ensembles, or any additional hyperparameters.

Our method works by learning a goal-conditioned value function via contrastive RL (CRL) (Eysenbach et al., 2022) and using that value function to train a goal-conditioned policy. The key ingredient is embarrassingly simple: when doing exploration, always condition the goal-conditioned policy on the single target goal. There are several intuitive reasons why we initially thought this method should work poorly:

  *(i)* Before the single target goal is reached, the value function will predict bogus values for that goal, so it should be unable to train the policy.

  *(ii)* There is no mechanism to drive exploration. Sampling a curriculum of goals that includes easy goals and leads to the target goal should perform much better.

This intuition turned out to be fallacious.

Empirically, we evaluate our approach on tasks ranging from bin picking to peg insertion to maze navigation, finding that it significantly outperforms prior methods that use a manually-designed curricula of subgoals (Eysenbach et al., 2022), methods that automatically propose subgoals for training (Chane-Sane et al., 2021), and even methods that use dense rewards (Haarnoja et al., 2018). While we do not claim that this is the best exploration method, it outperforms all alternative exploration strategies we have tried. Not only do we observe emergent skills during training, we find that different random seed initializations learn divergent strategies for solving the problem. While we still lack a theoretical understanding of *why* this approach is so effective, experiments highlight that *(1)* the contrastive representations used to express the value function are important, and that *(2)* the gains are not caused by "overfitting" the policy or the value function to the single target goal.

## 2 RELATED WORK

Our work builds on a long line of prior work in exploration methods for reinforcement learning (Jin et al., 2020; Thrun, 1992; Tokic, 2010; Stadie et al., 2015; Tang et al., 2017; Asmuth et al., 2009; Kearns & Singh, 2002; McGovern & Barto, 2001), and will study this problem in the specific setting of goal-conditioned RL (GCRL) (Kaelbling, 1993; Schaul et al., 2015; Andrychowicz et al., 2017; Lin et al., 2019; Eysenbach et al., 2021; Rudner et al., 2021; Savinov et al., 2018; Ding et al., 2019; Sun et al., 2019; Ghosh et al., 2020; Lynch et al., 2020; Dosovitskiy & Koltun, 2016; Schmeckpeper

---

[1]Some prior methods automatically propose training goals (Florensa et al., 2017; 2018; Sukhbaatar et al., 2018; OpenAI et al., 2021), yet these methods require additional machinery and are primarily evaluated on settings where subgoals lie on a 2-dimensional manifold.

et al., 2020; Nachum et al., 2018; Savinov et al., 2018; Srinivas et al., 2018; Nasiriany et al., 2019; Eysenbach et al., 2019). This section reviews three types of strategies for exploration. Our proposed method aims to lift the limitations associated with these prior methods.

**Rewards and demonstrations.** One of the key challenges with GCRL is the sparsity of the learning problem, so many prior GCRL methods assume access to a dense, hand-crafted reward function (Plappert et al., 2018a; Schaul et al., 2015) or a distance metric Trott et al. (2019); Tian et al. (2021); Hartikainen et al. (2020); Wu et al. (2019). Other methods attempt to make GCRL more tractable by using expert demonstrations Paul et al. (2019); Ding et al. (2019) to guide learning and planning. Although well-designed reward functions and expert demonstrations are useful for training, these components add complexity, and collecting demonstrations can be challenging. Our method builds upon a growing collection of GCRL algorithms that require neither a reward function nor demonstrations (Lin et al., 2019; Eysenbach et al., 2022; 2021; Zheng et al., 2024; Sun et al., 2019; Ghosh et al., 2020) – specifically, we consider a variant of GCRL where only a single goal is provided for training and evaluation. As such, we could treat it as a single-task problem, but we find that treating it as a multi-task problem is crucial to achieving good performance.

**Exploration and subgoal sampling.** Without a dense reward function or expert demonstrations, the primary challenge of GCRL is effective exploration. One class of exploration strategies adds noise to the actions (Fujimoto et al., 2018; Lillicrap et al., 2015; Heess et al., 2015; Haarnoja et al., 2018) or policies (Plappert et al., 2018b; Fortunato et al., 2018). While these methods are simple to implement, they typically fail to perform directed exploration (Osband et al., 2016a). A second class of methods formulates an intrinsic exploration reward (Machado et al., 2020; Pathak et al., 2017; Li et al., 2020; Eysenbach et al., 2018; Conti et al., 2018; Bougie & Ichise, 2020), which the agent aims to optimize in addition to the rewards provided by the environment. While these methods can work effectively, they can be challenging to scale to high-dimensional and long-horizon tasks. A third class of methods use probabilistic techniques, including ensembles (Osband et al., 2016a; Chen et al., 2017; Yao et al., 2021; Chen et al., 2018; Pearce et al., 2018), posterior sampling (Osband et al., 2016b; 2018; Dann et al., 2021; Fan & Ming, 2021), and uncertainty propagation (O'Donoghue et al., 2018; Tosatto et al., 2019) – these methods can excel at directed exploration, though challenges include tuning the prior and dealing with large ensembles. A fourth class of methods modifies the goals that are used in training. For example, some methods automatically propose subgoals, breaking down a hard task into a sequence of easier tasks (Chane-Sane et al., 2021; Zhang et al., 2022; Savinov et al., 2018; Shah et al., 2022; Zhang et al., 2024). We compare against one prototypical subgoal sampling method (RIS (Chane-Sane et al., 2021)). Other methods automatically adjust the goal distribution (Pong et al., 2020; Florensa et al., 2018; Venkattaramanujam et al., 2019) or initial state distribution (Florensa et al., 2017), so that the difficulty of learning increases throughout training. Despite excellent results in certain settings, scaling these methods beyond 2D navigation remains challenging, and the algorithms remain complex.

**Multi-task learning for single-task problems** The last strategy for exploration is so ubiquitous it is easy to forget: training on multiple related tasks, even when we only care about performance on a single difficult task. For example, many prior GCRL methods command a range of goals during exploration. Intuitively, the easy tasks can be learned with little exploration, and learning those tasks should enable the agent to solve more challenging tasks (similar to curriculum learning (Bengio et al., 2009; Matiisen et al., 2019; Campero et al., 2021)). However, actually constructing these multiple training tasks or goals requires additional human supervision: the human often lays out a "trail of breadcrumbs", and the agent learns how to navigate to each (Eysenbach et al., 2022). Our paper will study the setting where only a single goal is provided for training, yet our experiments will compare against baselines that have access to training goals with a range of difficulties.

## 3 SINGLE-GOAL EXPLORATION WITH CONTRASTIVE RL

### 3.1 PRELIMINARIES

**Notation.** We consider a controlled Markov process (i.e., an MDP without a reward function) defined by time-indexed states $s_t$ and actions $a_t$. Our experiments will use continuous states and actions. The initial state is sampled $s_0 \sim p_0(s_0)$ and subsequent states are sampled from the Markovian dynamics $s_{t+1} \sim p(s_{t+1} \mid s_t, a_t)$. Without loss of generality, we assume that episodes have an infinite horizon; finite horizon problems can be handled by augmenting the dynamics with an

absorbing state. We assume that the algorithm is given as input the single target goal state $s^*$ and aims to learn a policy $\pi(a_t \mid s_t)$ by interacting with the environment. *Unlike prior work, we do not assume that a distribution of goals for exploration is given; we do not assume that either a dense or sparse reward function is given.*

Following prior work (Eysenbach et al., 2021; Schroecker & Isbell, 2020), we define the objective as maximizing the probability of reaching the goal. Formally, define the $\gamma$-discounted state occupancy measure (Ho & Ermon, 2016; Syed et al., 2008; Dayan, 1993) as

$$\rho^\pi(s_f) \triangleq (1 - \gamma) \sum_{t=0}^\infty \gamma^t p_t^\pi(s_t = s_f), \tag{1}$$

where $p_t^\pi(s_t = s_f)$ is the probability of being at state $s_f$ at time step $t$. In continuous settings, $p_t^\pi(s_f)$ is a probability *density*. The objective is to find a policy that maximizes the likelihood of the single target goal under this occupancy measure:

$$\max_\pi \rho^\pi(s_f = s^*). \tag{2}$$

In discrete settings, this objective is equivalent to the standard discounted reward objective with a reward function $r(s_t, a_t) = \mathbb{1}(s_t = s^*)$; in continuous settings, it is equivalent to using a reward function $r(s_t, a_t) = p(s' = s^+ \mid s_t, a_t)$. Intuitively, this corresponds to maximizing the time spent at the goal. The hyperparameter $\gamma \in [0, 1)$ is part of the problem description.

**Contrastive RL.** Our method builds on *contrastive RL* (Eysenbach et al., 2022), prior work that uses temporal contrastive learning to solve goal-conditioned RL problems. This method was designed for a slightly different setting, where the input is a distribution over goals $p(g)$, and the aim was to learn a goal-conditioned policy $\pi(a \mid s, g)$ for reaching each of these goals. Contrastive RL is an actor-critic method. The critic $C(s, a, s_f)$ is learned so that it outputs the (relative) likelihood that an agent starting at state $s$ and taking action $a$ will visit state $s_f$. Following prior work, we parameterize the critic as the dot product between two learned representations, $\phi(s, a)^T \psi(s_f))$. The representations are not normalized. We will write the loss function in terms of these representations, which will turn out to be key for achieving good exploration.

To define the learning objective, we introduce a few distributions. Define $p(s, a)$ as the marginal distribution over state-action pairs in the replay buffer, and define $\rho(s_f \mid s, a)$ as the empirical discounted state occupancy measure, conditioned on a state $s$ and action $a$. Define $\rho(s_f)$ as the corresponding marginal distribution over future states. Contrastive RL uses these distributions to train the critic with a contrastive learning objective. Following prior work, we learn these representations using the infoNCE contrastive objective (Oord et al., 2018) together with a LogSumExp regularization that prior analysis (Eysenbach et al., 2022) has shown is necessary when using the infoNCE objective for control:

$$\max_{\phi(s,a),\psi(s_f)} \mathbb{E}_{\substack{(s,a)\sim p(s,a), s_f^{(1)} \sim \rho(s_f|s,a) \\ s_f^{(2:N)} \sim \rho(s_f)}} \left[ \underbrace{\log\left(\frac{e^{\phi(s,a)^T \psi(s_f^{(1)})}}{\sum_{j=1}^N e^{\phi(s,a)^T \psi(s_f^{(j)})}}\right)}_{\text{infoNCE}} - 0.01 \cdot \underbrace{\log\left(\sum_{j=1}^N e^{\phi(s,a)^T \psi(s_f^{(j)})}\right)^2}_{\text{LogSumExp regularization}} \right]. \tag{3}$$

In practice, this loss is implemented by sampling a random $(s_t, a_t)$ pair from the replay buffer and then sampling a future state $s_f = s_{t+\Delta}$ by looking $\Delta \sim \text{GEOM}(1 - \gamma)$ steps ahead. The negative examples are obtained by shuffling the future states (i.e., sampling from the product of two marginal distributions).

Once learned, the representations encode a Q-value (Eysenbach et al., 2022): $\phi(s, a)^T \psi(s_f) = \log Q(s, a, s_f) - \log \rho(s_f)$, where the Q value is defined with respect to the reward function introduced above. The policy is learned to maximize this (log) Q-value:

$$\max_\pi \mathbb{E}_{p(s)p(s_f)\pi(a|s,s_f)} \left[\phi(s, a)^T \psi(s_f) + \alpha \mathcal{H}(\pi(\cdot|s, s_f))\right], \tag{4}$$

where $\alpha$ is an adaptive entropy coefficient. Intuitively, the actor loss chooses the action $a$ that maximizes the alignment between $\phi(s, a)$ and $\psi(s^*)$.

---

**Algorithm 1 Single-goal Exploration with Contrastive RL.** The difference from most prior methods is that exploration is done by commanding a single difficult goal $s^*$, rather than sampling goals with a range of difficulties.

1: Initialize policy $\pi_\theta(a \mid s, g)$, replay buffer $\mathcal{B}$, classifier with logits $\phi(s, a)^T \psi(s_f)$.
2: **while** not converged **do**
3:     Collect one trajectory of experience using $\pi(a \mid s, s_f = s^*)$, add to buffer $\mathcal{B}$.
4:     Update representations $\phi(s, a), \psi(s_f)$ and policy $\pi(a \mid s, s_f)$ using contrastive RL.
5: Return policy $\pi(a \mid s, g = s^*)$.

---

## 3.2 OUR APPROACH

We now describe our approach to tackling the problem of learning to reach a single goal state. Our approach is a simple modification of contrastive RL: rather than asking the human user to provide many training goals for exploration, we always command the policy to collect data with the single hard goal $s^*$ (see Fig. 2). No other modifications are made to the algorithm. We use the same single goal state and success criteria as prior work (Yu et al., 2020); in each environment, the single goal corresponds to a state representative of successful task completion (e.g. closing a box or inserting a peg). The critic loss is the same as contrastive RL (Eq. 3). The actor loss is the same (Eq. 4). We will call this method "single [hard] goal CRL." Note that the single-goal in the algorithm name refers to the goal used for data-collection, not the goals used for policy updates. For training the actor, we use goals sampled from future states in the replay buffer. (Eq. 4 Ablation experiments (Fig. 10) show that only training the actor on the single goal decreases performance. We summarize the approach in Alg. 1.

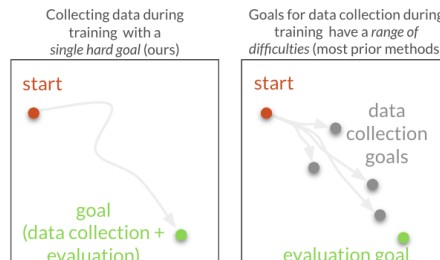

Figure 2: **Single-goal exploration.** *(Left)* Our method uses a single difficult goal for both data collection and evaluation. It is exceedingly unlikely that a random policy would ever reach this goal. *(Right)* Typical methods for goal-conditioned RL use a range of different goals for data collection, even if the user only cares about success at reaching a single difficult goal. These different goals can be provided by the user (Eysenbach et al., 2022) or generated with a GAN (Florensa et al., 2018), VAE (Nasiriany et al., 2019), or planning (Chane-Sane et al., 2021; Savinov et al., 2018; Zhang et al., 2022).

## 4 EXPERIMENTS

The main aim of our experiments is to evaluate the performance of single-goal contrastive RL compared to its multi-goal counterpart as well as prior baselines. We do so on four exploration-heavy, goal-reaching tasks, involving robotic manipulation and maze navigation. All experiments were run with five random seeds, and error bars in the plots depict the standard error. Hyperparameters can be found in Appendix B and code to reproduce our results is available online: `https://github.com/graliuce/sgcrl/tree/main`

**Tasks.** We measure the efficacy of our method on four goal-reaching tasks taken from prior work (Eysenbach et al., 2024), which are chosen to measure long horizon exploration. The tasks include three robotic manipulation environments (Yu et al., 2020) and one maze navigation environment (Eysenbach et al., 2019). The robotic manipulation tasks require controlling a sawyer robot to grasp an object (e.g., a block, lid, or peg) and accurately place it in a predetermined location (e.g., in a bin, on top of a box, or inside a hole). The point spiral task is a 2D maze navigation task. We quantify success in each episode by whether the agent reached sufficiently close to the goal in at least one state in an episode. This 0/1 sparse reward signal is used by a few of our baselines but is not needed by contrastive RL. Note that while (Eysenbach et al., 2019) measured success by evaluating on goals with a range of difficulties, we evaluate only on the single, hard goal, which corresponds to successful task completion in the Metaworld benchmark (Yu et al., 2020).

These tasks present exceedingly difficult exploration challenges. To quantify the difficulty, we measured the success rate under a uniform random policy: for each of the sawyer environments, we did not see a single success state in 75,000,000 environment steps (600,000 episodes); for the point spiral environment, we did not see a single success state in 40,000,000 environment steps (400,000 episodes). Previous methods solve these tasks with the aid of dense rewards (Yu et al., 2020), or

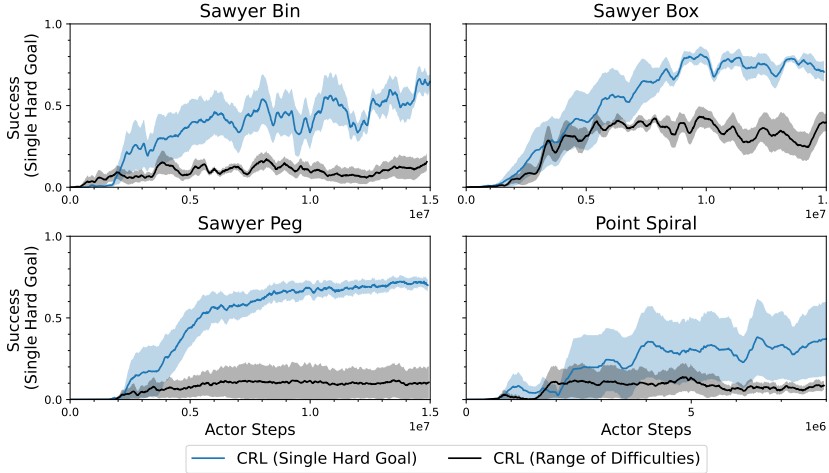

Figure 3: **Single goal Exploration is Highly Effective.** We compare single hard goal exploration (command the single hard goal in every trial) to "range of difficulties" exploration (sampling uniformly from a human-provided set of easy/medium/hard goals). In each of the four environments, single-goal exploration yields considerably higher success rates, all while being easier for the human user.

with a rich curriculum of subgoals (Eysenbach et al., 2019), but the single-goal CRL agents must accomplish these tasks with no hand-crafted rewards, demonstrations, or subgoals.

## 4.1 SINGLE-GOAL EXPLORATION IS EXCEEDINGLY EFFECTIVE.

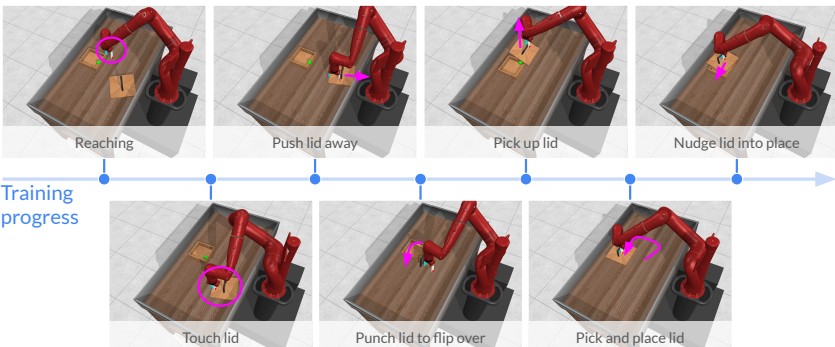

Figure 4: **Skills and Directed Exploration for Putting a Lid on a Box:** This manipulation task contains an open box and a lid. The single fixed goal has the lid placed neatly on top of the center of the box. The images above show skills acquired throughout the course of learning. Note that some skills unlock subsequent skills (e.g., reaching is a prerequisite for picking, which is a prerequisite for placing) while others look like open-ended "play" (flipping the lid over, pushing the lid away from the box).

**A single goal works well.** *We find that Contrastive RL effectively solves these four tasks*: equipped only with a single target goal, the agent automatically explores the environments and learns complex manipulation skills (see Fig. 3). We compare this method to an "oracle" variant that is trained on human-designed goals that vary in difficulty, ranging from easy goals to the single hard goal. Surprisingly, our proposed method significantly outperforms this "range of difficulties" method.

While we have never seen a random policy solve any of these tasks, our method achieves its first success within thousands of trials: 3,197 trials for `sawyer box`, 9,329 trials for `sawyer bin`, 15,895 trials for `sawyer peg`, and 11,724 trials for the spiral task.[2]

---

[2]These numbers are averaged across the random seeds.

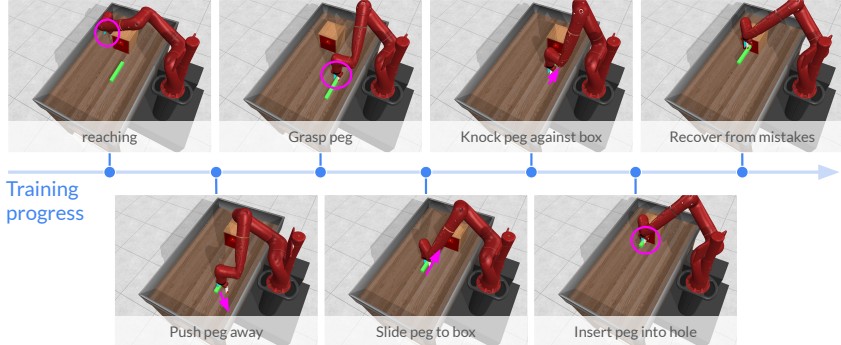

Figure 5: **Skills and Directed Exploration for Peg Insertion:** This manipulation task contains a peg and box with a narrow hole; the single fixed goal is a state where the peg is inside the hole. The agent acquires a sequence of increasingly complex skills throughout training, some of which are important for solving the task (e.g., reaching, grasping) while others are more "playful" (e.g., knocking the peg against the box). The agent also learns to recover from mistakes (see Fig. 8).

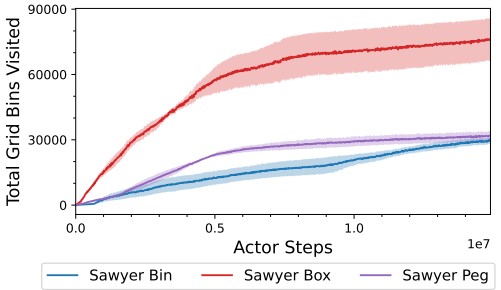

Figure 6: **Quantifying Exploration.** We analyze "single hard goal" exploration by discretizing the object's XY position and counting the cumulative number of unique positions visited throughout training. Exploration starts to plateau after the agent can successfully reach the goal (compare with Fig. 3).

(a) "Push and flick" (seed = 3)  (b) "gently pick" (seed = 4)

Figure 7: **Different random seeds learn different strategies.** Other seeds learn a policy whose strategy depends on the initial position of the block.

**Early training: agent develops an emergent curriculum of skills.** Not only does single-goal CRL consistently achieve high success rates on manipulation tasks, but it also demonstrates complex and directed exploration techniques during early training. To observe the agent's behavior throughout training, we saved learning checkpoints at fixed intervals and visualized the agent's behavior at each checkpoint (see Figures 1, 4 and 5). We found that the agent learns simple skills before complex ones. For example, in all three environments, the agent *(1)* first learns how to move its end-effector to varying locations, *(2)* then learns how to nudge and slide the object, and *(3)* finally learns to pick up and direct the object. We observe a wide array of exploratory behavior in early training that is not directly connected to the goal: from punching the box lid to flip it over (Fig. 4) to pushing the block far away in a random direction (Fig. 1).

**First successes: agent trades off exploration for exploitation.** As the agent learns to reach the goal more consistently, the agent's behavior becomes less exploratory, qualitatively similar to UCB exploration (Guo et al., 2020; Osband et al., 2016b; 2018; Dann et al., 2021; Fan & Ming, 2021). To quantify exploration, we discretized the state space of the robotic manipulation environments and recorded the cumulative number of unique positions visited throughout training. Fig. 6 shows that the growth rate of this exploration metric decreases as the success rate increases (compare with Fig. 3). For the `sawyer box` and `sawyer peg` environments, the agent achieves a high success rate earlier in training, which corresponds to the earlier plateau of the unique grid cells curve. For the `sawyer bin` environment, the agent takes longer to reach high success, and the exploration metric does not start leveling out until the end of training. This trend highlights how single-goal CRL develops a self-directed exploration strategy that automatically trades off between exploration and exploitation.

**Consistent successes: agent finds diverse paths to the goal.** Not only is the performance of single-goal CRL reproducible (all random seeds solve the manipulation tasks), but policies trained

with different random seeds learn qualitatively different goal-reaching strategies. For example, Fig. 7 shows the strategies learned by different random seeds on the `sawyer bin` task. One seed consistently moves the block flush against the wall of the red bin and flicks it into the blue bin. Another seed tends to grasp the block, lift it, and gently drop it in the blue bin. A third seed chooses between these strategies depending on whether the block starts near the wall or away from the wall. Without explicit human guidance in the form of rewards, demonstrations, or subgoals, we see multiple creative and divergent strategies emerge for solving the same problem.

**Further training: agent develops robustness and self-recovery.**   During later stages of training, we observe that the agent demonstrates robustness and learns to recover from mistakes. For example, in the `sawyer peg` environment, when the agent drops the peg, it is able to recover by bending down and grasping the peg again. To quantify robustness, we ran perturbation experiments in the `sawyer peg` and `sawyer box` environments, in which we randomly perturbed the target object's location between 0 and 0.05 meters along each of the three axes. We tested two settings: *(1)* perturbation at the start of the episode ("static perturbations") and *(2)* in the middle of an episode ("dynamic perturbations": $t = 20$ for `sawyer box` and $t = 50$ for `sawyer peg`). As shown in Fig. 8, single goal exploration is robust to static perturbations and somewhat robust to dynamic perturbations, notably outperforming multi-goal CRL in three out of four scenarios. We hypothesize that this is also the result of better exploration, which leads to learned representations that generalize better across unusual or unseen states.

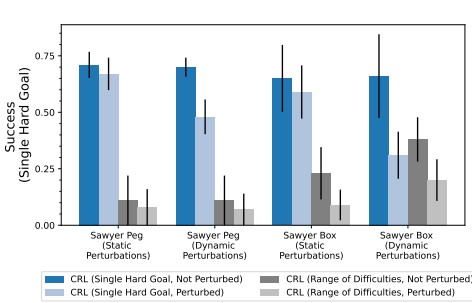

Figure 8: **Robustness to perturbations:** Single (Hard) Goal exploration results in policies that are more robust to environment perturbations, as compared to policies trained with goals ranging in difficulty. The success rate remains high even when the object is perturbed at the start ("static") or in the middle ("dynamic") of an episode, likely because its effective exploration means that it has seen a wide range of states during training.

## 4.2   The RL Algorithm is Key

We compare against a number of algorithms to investigate the importance of the underlying RL algorithm: is single goal exploration useful for other goal-conditioned RL algorithms?

**Baselines.**   We compare single-goal CRL against prior methods that aim to address the sparse reward problem by making additional assumptions or employing additional machinery for exploration. Reinforcement learning with imagined subgoals (RIS) (Chane-Sane et al., 2021)maintains a high-level policy that predicts subgoals halfway to the end goal and learns the behavioral policy to reach both the subgoal and the end goal. We also employ a few variants of Soft Actor-Critic (SAC) with additional assumptions: SAC with sparse rewards, SAC+HER with sparse rewards, and SAC with dense rewards. In the sparse rewards setting, the agent receives a reward of 1 near the goal and 0 otherwise. In the dense reward setting, the agent receives a continuous reward tailored to the environment, using the distance to the goal for the spiral task and the Metaworld (Yu et al., 2020) reward function for sawyer tasks. Table 1 summarizes the assumptions for each method.

**Results.**   As shown in Fig. 9 single-goal contrastive RL significantly outperforms these alternative methods, showing that the underlying RL algorithm is important for single goal exploration. Notably, prior methods rarely reach the goal at all, with the exception of RIS on the simplest task (`point spiral`). To verify that our implementation of SAC was correct, we also plotted the reward function throughout training (recall that SAC has access to a reward function, while other methods do not) and observed that it increases throughout the course of learning.

## 4.3   Why does Single-Goal Exploration Work?

In this section, we present ablation experiments that provide insight into the workings of single-goal CRL. We find that using single-goal data collection and an inner product critic are both key to the

Table 1: **Baselines:** Assumptions for methods used in the experiments below.

| Algorithm | Exploration | Requirements | | |
| | | Exploration Goals | Dense Rewards | Distance Threshold |
| --- | --- | --- | --- | --- |
| Contrastive RL | single goal (ours) | ✗ | ✗ | ✗ |
| | multiple goals | ✓ | ✗ | ✗ |
| SAC (sparse rewards) | single goal | ✗ | ✗ | ✓ |
| SAC (dense rewards) | single goal | ✗ | ✓ | ✓ |
| SAC (sparse rewards) + HER | single goal | ✗ | ✗ | ✓ |
| RIS | single goal | ✗ | ✗ | ✓ |

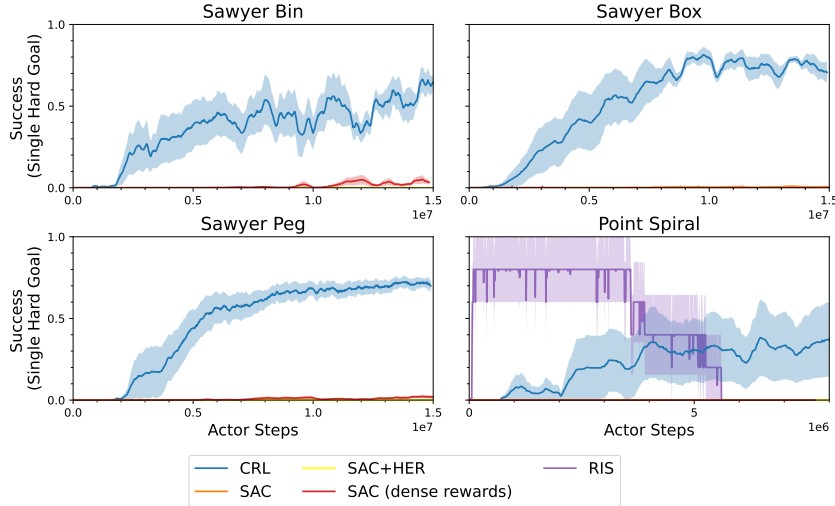

Figure 9: **The RL Algorithm Matters.** We compare several underlying RL algorithms all using single-goal exploration, with Table 1 highlighting the assumptions of each method. CRL outperforms these prior methods, showing that *(i)* single-goal exploration is only effective with the right underlying RL algorithm and that *(ii)* with this algorithm, we can achieve considerably higher performance while making fewer assumptions (no rewards).

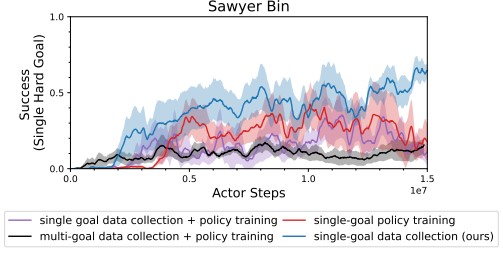

Figure 10: **Ablation experiments.** While our method uses an actor loss that uses many goals (Eq. 4), alternatives that train the policy with a single goal perform no better.

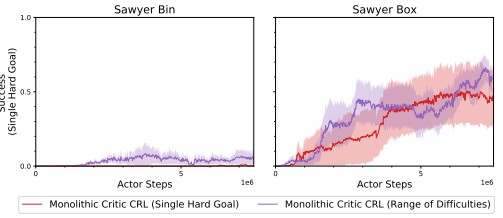

Figure 11: **The importance of representations.** Single goal exploration is less effective with using a monolithic critic architecture (as opposed to an inner product architecture), suggesting that the contrastive representations may drive exploration.

effectiveness of the method. Additionally, the performance boost of single-goal data collection does not seem to be due to overfitting of the policy parameters on the task.

**The effectiveness of single-goal exploration is not explained by overfitting.** One possible explanation for the method's success is that the algorithm overfits its policy parameters on the single-goal task. That is, if the agent only collects data conditioned on the single goal, the algorithm does not just overfit the policy parameters to reach the single goal, but also learns useful representations for states along the path from the starting point to the goal. To test this hypothesis, we compared our method (which randomly samples different goals (see Eq. 4)) with a variant that always uses the single hard goal in the actor loss. Note that this single hard goal is the one that is used for evaluation. If overfitting were occurring, we would expect that modifying the actor loss to only train on the single goal would boost performance. However, the results in Fig. 10 show that this is not the case.

When data are collected with a single goal, using a single goal in the actor loss degrades performance (purple vs. blue (ours)). When data are collected with multiple goals, using a single goal in the actor loss gives only a slight boost performance (red vs. black). In short, the performance of single goal exploration is not explained by overfitting.

**Representations are important.** Our next set of experiments study how the representations might drive exploration. To do this, we replaced the inner product critic function ($\phi(s,a)^T \psi(s_f)$) with a monolithic critic function ($Q(s,a,s_f)$), which takes as input a concatenated array of the state, action, and goal.[3] The results, shown in Fig. 11, show that single-goal exploration is not effective with using a monolithic critic network. We found that using this modified critic function resulted no performance boost for the single-goal strategy, and in fact the single-goal method performed worse than the multi-goal method. This experiment suggests that the contrastive representations $\phi(s,a)^T$ and $\psi(s_f)$ could be important in driving single-goal exploration, though the precise mechanism for *how* they drive exploration remains unclear. In summary, single-goal exploration is only effective when combining the right algorithm (CRL) with the right critic architecture. In Appendix A, we show that environment dynamics are reflected in the contrastive representations early in training.

## 5 CONCLUSION

In this paper, we showed that skills and directed exploration emerges from a straightforward RL algorithm: contrastive RL where every trajectory is collected by trying to reach a single fixed goal. The resulting method has many appealing properties of prior exploration methods: in each episode the agent seems to try to visit some new state or attempt some new behavior. We do not see the random dithering that is common with naïve exploration methods like $\epsilon$-greedy. Moreover, this method does not require *any* additional hyperparameters: this is in contrast to even the simplest exploration algorithms today, which require a scale parameter (e.g., Gaussian noise in TD3 (Fujimoto et al., 2018)). And, while prior exploration algorithms include a schedule (more hyperparameters!) for gradually decreasing the degree of exploration throughout the course of learning, such behavior emerges automatically from our proposed method.

There remain two important outstanding questions raised by our experiments. *First*, we lack a clear understanding of why skills and directed exploration emerge. Our experiments provide some hints (representations are important; it is not explained by overfitting), yet much theoretical work remains to be done to understand exactly what is driving the exploration. A rich theoretical understanding of the mechanisms driving the exploration here is important not only for explaining the success of this method, but it may also provide insights into how to adapt the method here to other settings. *Second*, how can we leverage the success of the proposed method to address exploration in other problem settings (e.g., if a reward function were given or if not even a single fixed goal were available). The "impossible goal" experiment in Appendix C provides one simple approach, but there likely exist significantly better methods of exploiting the emergent exploration properties that we have observed in the single-goal setting.

The emergence of divergent strategies for solving the bin picking task points towards broader research questions about autonomous capabilities. The RL research community has built much scaffolding to help agents succeed. However, we have shown that without this scaffolding, agents develop unexpected and unique methods for problem-solving. Perhaps this seemingly creative behavior emerges not in spite of but because of the lack of human guidance. We encourage future research to explore what RL can accomplish in the absence of human intervention.

**Limitations.** The primary limitation of our work is a lack of theoretical analysis explaining *why* skills and directed exploration emerge. Empirically, our experiments are focused primarily on manipulation tasks; we encourage future work to study applications to other settings.

**Reproducibility Statement.** We provide the source code (`https://github.com/graliuce/sgcrl/tree/main`) for reproducing the primary results of the paper, shown in Figure 3. The codebase supports experiments on all four environments, for both the single-goal and

---

[3]In this experiment only, we decreased the batch size from 256 to 32 for both methods, as otherwise we encountered out-of-memory errors for the monolithic method.

multi-goal versions of the CRL algorithm. The hyperparameters used for the CRL algorithm and baselines are given in Appendix B.

**Acknowledgments.** Thanks to the members of the Princeton RL lab for feedback on preliminary versions of the work. We are also grateful to Elliot Chane-Sane for corresponding with us on the strengths and limitations of the RIS algorithm. The experiments in this paper were substantially performed using the Princeton Research Computing resources at Princeton University, which is a consortium of groups led by the Princeton Institute for Computational Science and Engineering (PICSciE) and Office of Information Technology's Research Computing.

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

## A  ENVIRONMENT DYNAMICS ARE VISIBLE IN THE NORMS OF CONTRASTIVE REPRESENTATIONS

To further investigate our predictions about targeted exploration and robustness arising from the building of rich representations early in training, we visualized the norms of the contrastive goal encoder $||\psi(s_g)||_2^2$ at an early checkpoint, shown in Figure 12. We target goal encoder norms since we observe that mean goal encoder norms over the training distribution rollout positions closely correlate with training loss, and hypothesize that these norms reflect environment-learning.

Specifically, we fix the end-effector distance to the target distance for gripping, uniformly randomly sample many states $x_i$ corresponding to both the agent end-effector and object being at $x_i$, and plot the corresponding values $||\psi(x_i)||_2^2$ from an early (pre-first-success) encoder checkpoint on the `sawyer bin` task.

As shown in Figure 12, we find interpretable patterns corresponding to a map of environment dynamics, such as high goal encoder norms at the location of the impassable bin walls and bin bottom. We hypothesize that these strongly-represented environment features contribute to the development of higher-level skills and ultimately behaviors such as perturbation robustness.

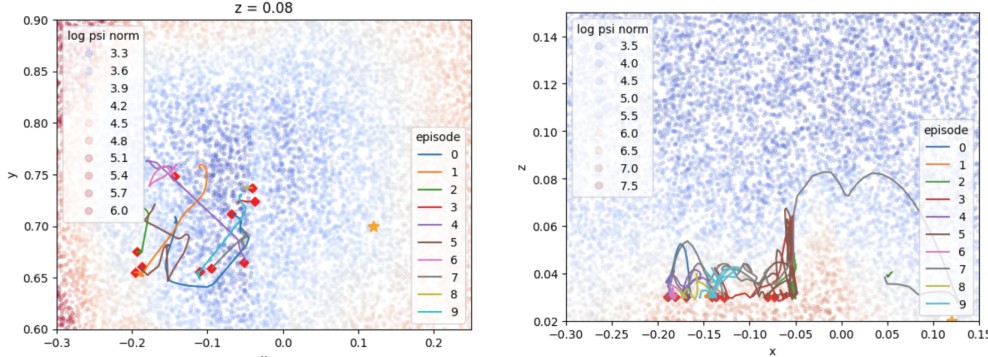

Figure 12: **Contrastive representations capture environment dynamics (impassable bin walls and floor) in an interpretable way**. We plot the log of the goal encoder norms, $\log(||\psi(s_g)||_2^2)$, for various observations where the end effector and block are at the same location. We show a top-down view at box-height (left) and side view (right). We find that the impassable bin wall is visible both from the top view (thin strip of lighter blue) and side view (vertical spike of red), represented by relatively higher norms. Example policy rollouts (colored lines) from checkpoints at the same stage in training are overlaid for reference, with their starting positions marked as red diamonds and the goal marked as a gold star.

# B    EXPERIMENTAL DETAILS

Table 2: Hyperparameters for our method and the baselines.

| hyperparameter | value |
|---|---|
| Contrastive RL (CRL) (Eysenbach et al., 2022) | |
| batch size | 256 |
| learning rate | 3e-4 |
| discount | 0.99 |
| actor target entropy | 0 |
| hidden layers sizes (policy, critic, representations) | (256, 256) |
| initial random data collection | 10,000 transitions |
| replay buffer size | 1e6 |
| samples per insert[1] | 256 |
| representation dimension ($\dim(\phi(s, a))$, $\dim(\psi(s_g))$) | 64 |
| actor minimum std dev | 1e-6 |
| SAC (Haarnoja et al., 2018) | |
| batch size | 256 |
| learning rate | 3e-4 |
| discount | 0.99 |
| hidden layers sizes (policy, critic) | (256, 256) |
| target EMA term | 5e-3 |
| initial random data collection | 10,000 transitions |
| replay buffer size | 1e6 |
| samples per insert[1] | 256 |
| actor minimum std dev | 1e-6 |
| RIS (Chane-Sane et al., 2021) | |
| batch size | 256 |
| learning rate | 1e-3 (critic), 1e-4 (policy) |
| high-level policy learning rate | 1e-4 |
| discount | 0.99 |
| hidden layers sizes (policy, high-level policy, critic) | (256, 256) |
| initial random data collection | 10,000 transitions |
| replay buffer size | 1e5 |
| Polyak coefficient for target networks | 5e-3 |
| valid state KL constraint ($\epsilon$) | 1e-4 |
| subgoal KL penalty ($\alpha$) | 0.1 |
| high-level policy weight regularization ($\lambda$) | 0.1 |

[1] How many times is each transition used for training before being discarded.

[2] We collect $N$ transitions, add them to the buffer, and then do $N$ gradient steps using the experience sampled randomly from the buffer.

## C   IMPOSSIBLE GOALS

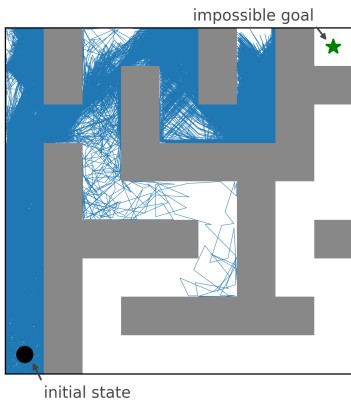

Figure 13: **Single-goal exploration with an impossible goal.**

### C.1   IMPOSSIBLE GOALS.

To further probe single goal exploration, we tried commanding a goal that was impossible to reach in a maze navigation task. One might expect to see completely random behavior, or see no behavior at all. Visualizing the trajectories visited throughout training (Fig. 13), we observe that the agent seems to try to navigate to that impossible goal but then gets stuck. This pattern suggests that commanding an impossible goal is a plausible strategy for improving exploration. However, it fails to explore a fair number of states, suggesting that there are likely more effective ways of inducing exploration in settings without a single fixed goal.

# D    VISUALIZING EMERGENT EXPLORATION IN POLICY ROLLOUTS

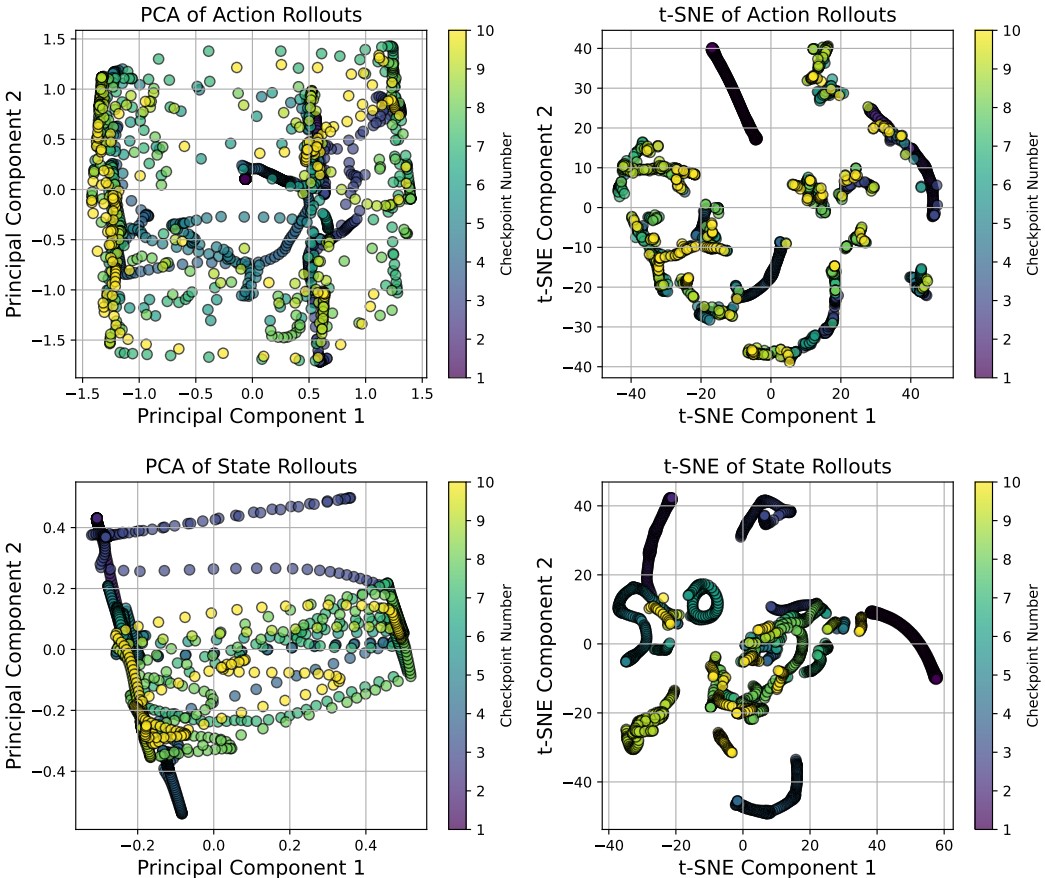

Figure 14: **Rollouts at early checkpoints indicate distinct exploratory behavior**. Here we plot action and state rollouts after PCA and t-SNE dimensionality reduction for the first 10 checkpoints of the model on the Sawyer Peg environment.

To further analyze the progression of exploration and skill learning, we apply dimensionality reduction on the states encountered and the actions taken for one episode at the first 10 checkpoints of the single-goal CRL model on the sawyer peg environment. The checkpoints were saved at fixed time intervals (e.g. every 15 minutes), which corresponded to 3,300 trials in the environment. We observe that the action and state rollouts for early checkpoints (2,3,4) tend to be more spatially separated in the PCA and T-SNE plots (Fig. 14), indicating unique exploratory behavior, whereas the rollouts for later checkpoints (8,9,10) have more overlap, indicating the algorithm has converged to similar behavior patterns once the single-goal has been reached.

# E EVALUATION OF SINGLE-GOAL METHOD ON MULTIPLE GOALS

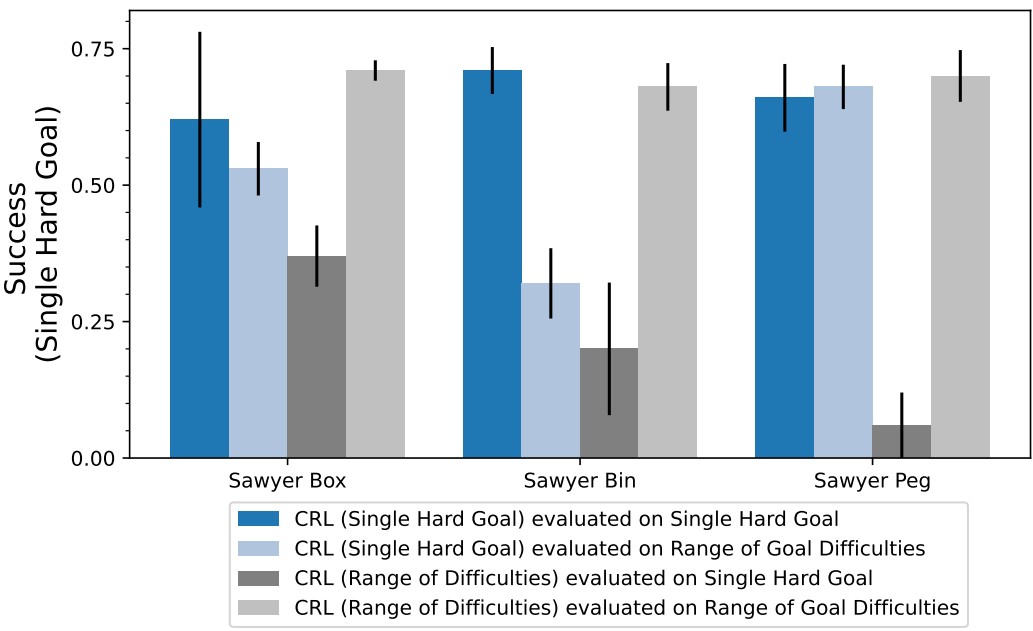

Figure 15: **Generalization to multiple goals.** Single-goal CRL can still reach goals along the path from the starting point to the goal.

We evaluate single-goal CRL and multi-goal CRL on the distribution of goals used for data collection in the multi-goal algorithm in addition to the single, hard goal used for evaluation in the main text. Figure 15 shows that single-goal CRL retains the ability to reach these goals (i.e. accomplish multiple tasks) without being explicitly guided to learn those tasks. The finding is most prominent in the Sawyer Box and Sawyer Peg tasks. This evaluation, in combination with the ablation experiment (see Fig. 10), suggests that single-goal CRL is not overfitting its policy parameters on a single task. Instead it performs multi-task learning, which surprisingly improves performance on the single task as well compared to prior methods.

# F  SENSITIVITY OF REPRESENTATIONS TO STATE DIMENSIONS

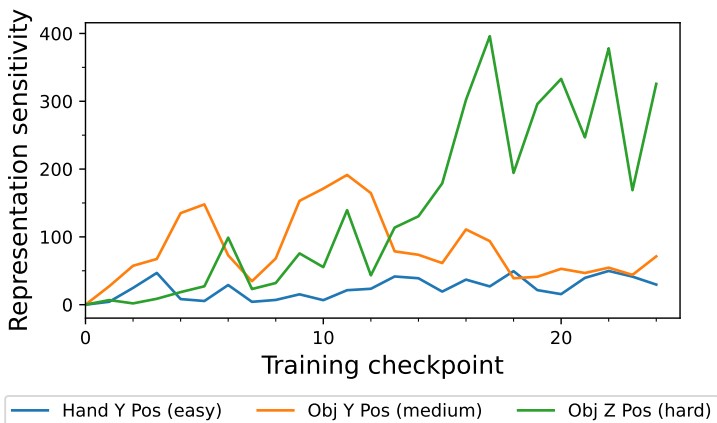

Figure 16: **Representation sensitivity to state dimensions evolves throughout training.** We probe single-goal CRL representations trained on the Sawyer Box environment for sensitivity to three state dimensions - the hand y position, the object y position, and the object z position (height). For each checkpoint, we measure the representation sensitivity by the l2 distance between the initial state representation before $(s_0^1)$ and after $(s_0^2)$ a small perturbation of 0.1 to the given state dimension: $||\phi(s_0^1, a_0) - \phi(s_0^2, a_0)||_2$.

We further probe the single-goal CRL representation for sensitivity to three state dimensions in the sawyer box environment: The y position of the hand (end-effector), the y position of the object (box lid position on table), and the z position of the object (box lid height). These dimensions were chosen because changing these dimensions requires increasing levels of skill; moving the end-effector y position is easy, moving the object y position is medium difficulty, and changing the object height is hard. For each training checkpoint, we measure the representation sensitivity by the l2 distance between the initial state representation before $(s_0^1)$ and after $(s_0^2)$ a small perturbation of 0.1 to the given state dimension: $||\phi(s_0^1, a_0) - \phi(s_0^2, a_0)||_2$. We find that the representations are more sensitive to the object y position during earlier stages of training and more sensitive to the object z position during later stages of training. This shift in sensitivity to the difficulty of the state dimension suggests that the method automatically develops a reasonable curriculum for exploration.

# G    TRANSFERRING REPRESENTATIONS BETWEEN TASKS

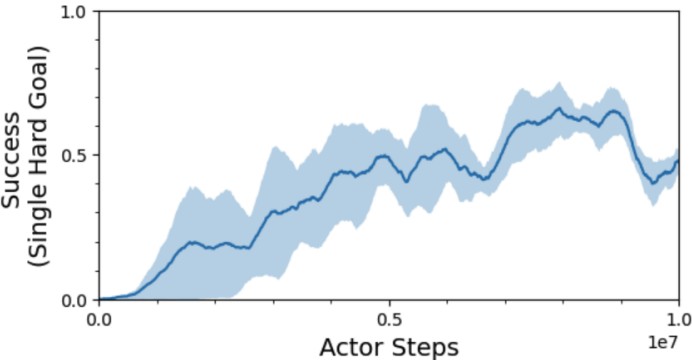

Figure 17: **Transferring representations between tasks may improve sample efficiency**. After training the single-goal CRL representation for 15M environment steps to place a block in a box, we then use these representations to initialize an agent for solving the Sawyer Bin task.

In this experiment, we evaluate whether transferring single-goal CRL representations between tasks can improve training sample efficiency. Because single-goal CRL representations learn interpretable environment dynamics (see Fig. 12), one could expect these learned properties to transfer between tasks. Over three random seeds, we train CRL representations for 15M environment interactions on a task that involves placing a block in a box. We then use these representations to initialize an agent for solving the Sawyer Bin task. We find that this process leads to better sample efficiency (compare Fig. 17 and Fig. 3 top left), though the difference is not drastic. Overall, this experiment provides some preliminary indication that single-goal CRL representations are transferrable between tasks.

## H  SINGLE-GOAL AND MULTI-GOAL CRL SHOW SIMILAR PERFORMANCE ON AN EASY GOAL

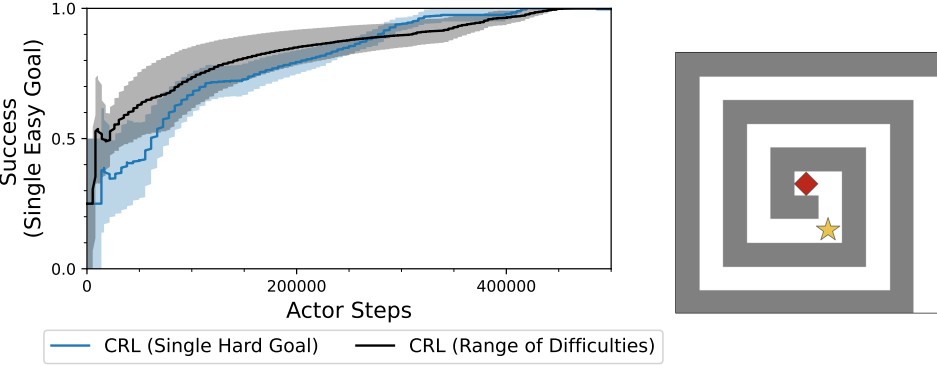

Figure 18: **Single-goal and multi-goal CRL show similar performance on an easy goal**. We train multi-goal CRL and single-goal CRL on the Point Spiral environment but set the single goal for single-goal CRL to be an easy goal close to the starting state. The starting position is marked as the red diamond and the single, easy goal is marked as a gold star.

To verify our implementation of multi-goal CRL, run an experiment where we use a single, easy goal instead of a single, hard goal in the Point Spiral environment. We observe that single-goal and multi-goal CRL show similar training progress and both are able to completely solve the environment (always reach the easy goal) within 500,000 actor steps. This experiment suggests that the performance difference that we observe between single-goal CRL and the other methods is due to the difficulty of the given task.

