# OpenReview forum: "A Single Goal is All You Need: Skills and Exploration Emerge from Contrastive RL without Rewards, Demonstrations, or Subgoals"
_ICLR.cc/2025/Conference — ICLR 2025 Poster_

### Official Review · Reviewer_gayD · 2024-10-27

**Soundness:** 3
**Presentation:** 3
**Contribution:** 3
**Rating:** 8
**Confidence:** 4

**Summary:**

This work builds upon previous work in contrastive reinforcement learning. What's unique to this work is how their learning algorithm conditions on a single goal; this has implications on the distribution of data within the replay buffer which ultimately impacts the learning dynamics and exploration behaviour of the agent. The authors argue that this approach results in emergent skill learning without explicitly having to define a reward function or learning curriculum for the task being solved. This argument is supported with simulated experiments on robotic manipulation and maze traversing benchmarks. The performance of their approach is compared to contrastive reinforcement learning using a range of human-specified goal difficulties and existing approaches (SAC+HER and RIS). The authors justify their claim with empirical results, leading to the insight that contrastive reinforcement learning with conditioning on a single goal demonstrates advantageous exploration and skill learning behaviours relative to existing approaches that seek to learn effective exploration and skill learning strategies.

**Strengths:**

- The authors provide valuable insights into the effectiveness of training a policy conditioned on a single goal with contrastive reinforcement learning in the single-task setting.

- The authors contribute to the discussion on skill learning through empirically demonstrating the emergence of skill learning with contrastive objective functions.

- The authors validate each of their claims and provide discussions on counter arguments (e.g. overfitting single-task) that support their empirical results.

- The authors provide sufficient information and high-quality materials to reproduce their results.

**Weaknesses:**

- There could be more analysis of emergent skill learning, in the current draft the authors mention that fixing a checkpoint and examining the qualitative behaviour of the policy they see evidence of skill learning, further justification of this claim is required in my opinion.

- The authors highlight an awareness of this point but further analysis on why this strategy works would be useful.

- It would be interesting to see if this approach results in exploration strategies that are capable of solving more complex tasks. For example tasks that encounter Sussman's anomaly, where naive subgoal design may not work well. I am unsure how this approach would perform but it would be promising if it did well and might further justify the strength of this more general approach to learning to solve tasks with reinforcement learning. Also tasks with significant bottlenecks in which naive exploration strategies are often incredible sample inefficient, does this approach help improve performance on such tasks?

- While the authors address the concern of overfitting to the individual task being solved, they do not explore how well the learnt representations generalise. While there are examples of solving tasks with minor perturbations there isn't a discussion of more significant changes to the task environment. From a practical standpoint, it would be interesting to understand how well this approach can be used to learn value representations that generalise well across task environments.

- This work suffers from ailments of reinforcement learning approaches more generally, most especially sample efficiency. It would be interesting to understand if the representations learnt in this work can be leveraged to address the issue of sample efficiency when learning multiple tasks (e.g. on the metaworld benchmark).

**Questions:**

Thank you for contributing this work. When reading the paper the emergent skill learning argument felt like it needed further justification.


(1) I wished to clarify if the following pattern of skill emergence always held? The paper suggests yes but I wished to confirm this point.

"For example, in all three environments, the agent (1) first learns how to move its end-effector to
varying locations, (2) then learns how to nudge and slide the object, and (3) finally learns to pick up
and direct the object. "

(2) I also wished to ask when you mention fixed sampling intervals for checkpoints, what fixed interval was used? Were there any exceptions to the skill learning trends identified in (1) (e.g. checkpoints with purely random behaviours) and if so how frequent were exceptions?

In answering (1) and (2) it would be useful to provide:

(a) Video's indicating the progression of skill learning with details of the checkpoints, this would be helpful for validating the qualitative behaviours claimed within the paper.

(b) Not entirely necessary but it would be nice to have quantitative metrics for characterising the various skills. For instance when you mention learning how to move the end-effector, distinguishing between purely random movements and directed movement would be useful, there are certain quantitative metrics that could be used, such as correlations between actions. Contrasting this with naive exploration approaches such as SAC may yield insights and further validate the skill learning claim. Similarly for learning to interact with objects and eventually pick them, characterising these skills quantitatively would be useful.

(c) It may be possible to perform clustering on policy rollouts data to identify distinct exploratory behaviours as learning progresses. This would be a valuable addition.

---

> ### Author Response · Authors · 2024-11-17
> **Author Response**
>
> Dear Reviewer,
>
> Thank you for the time and effort put into this review, as well as the suggestions for improving the paper. It seems that the main inquiries relate to further analysis of skill learning. To address this concern, we have added new analysis on policy rollouts (see new Fig. 15) as suggested by the reviewer. We have also highlighted unique skill learning in a new, highly-difficult environment (see new Fig. 14) and added additional analysis on representation probing (see new Fig. 17). **Do the new analysis and experiments address the reviewer's concerns about the paper?** We look forward to continuing the discussion!
>
> > analysis of emergent skill learning:  perform clustering on policy rollouts data to identify distinct exploratory behaviours as learning progresses.
>
> Thank you for the suggestion! We have performed this clustering and added the visualization to Appendix D (see new Fig. 15). We observe that the action and state rollouts for early checkpoints (2,3,4) tend to be more spatially separated in the PCA and T-SNE plots, indicating unique exploratory behavior and skill learning, whereas the rollouts for later checkpoints (8,9,10) have more overlap, indicating the agent has converged to similar behavior patterns once the single-goal has been reached.
>
> > Would like to see further analysis on why this strategy works
>
> As suggested by the reviewer, we did further analysis of the representations to understand what state features they were picking up on, and how this changed throughout the course of learning. We find that the representations are more sensitive to perturbations along “easier” state dimensions during early training and more sensitive to perturbations in “harder” state dimensions during later training stages (see new Fig. 17 in Appendix F), suggesting that the method automatically develops a reasonable training curriculum.
>
> While our paper is empirical, we are also very interested in, and have started investigating, the possible theoretical reasons why this method works so well. We speculate that there might be something to do with the connection between CRL and the graph Laplacian [1, 2], which could allow CRL representations to capture the large-scale geometry of the environment and drive directed exploration [3]. If the reviewer has suggestions on theoretical approaches, we'd love to continue the discussion!
>
> > Would be interesting to see if this approach results in exploration strategies that are capable of solving more complex tasks (where naive subgoal design may not work well)
>
> Thank you for the suggestion! In addition to the current tasks, which are already quite challenging (see SAC+HER and RIS in Fig. 9), we have added a new environment (sawyer box hard) in which the agent must place a block in the box and then place the lid on top of the box. This environment is especially difficult because the required behavior is non-linear and requires multi-step planning. While single-goal CRL was not able to consistently solve this environment (success rate < 0.05), we observed unexpected emergent skills during training which were not observed in the simpler environments. We showcase a few of these emergent skills in a [video uploaded to the anonymous code repository](https://anonymous.4open.science/r/sgcrl-C100/sawyer_box_hard.mp4), and we have added a section about this environment in Appendix C (see new Fig. 14). We believe that this behavior further supports our main contribution that single-goal CRL drives exploration and skill learning, though it has not solved this difficult, multi-step task.
>
> > It would be interesting to understand how well this approach can be used to learn value representations that generalize well across task environments … Can the learned representations be leveraged to address the issue of sample efficiency when learning multiple tasks?
>
> To address this question, we are currently running an experiment where we train the agent using single-goal CRL to place a block in the box. Taking that trained model, we will then train it on the sawyer box (box closing) task and observe whether there is improvement in sample efficiency. If so, this experiment would indicate that skills learned by single-goal CRL are transferable to new tasks.

---

> > ### Author Response · Authors · 2024-11-18
> > **Results of Representation Transfer Experiment**
> >
> > >It would be interesting to understand how well this approach can be used to learn value representations that generalize well across task environments … Can the learned representations be leveraged to address the issue of sample efficiency when learning multiple tasks?
> >
> > The experiment has finished running and we have added the results to Appendix G (see new Fig. 18). In this experiment, we evaluate whether transferring single-goal CRL representations between tasks can improve training sample efficiency. Over three random seeds, we train single-goal CRL representations on a task that requires the agent to place a block in a box. We then use these representations to initialize an agent for solving the Sawyer Bin task (we chose Sawyer Bin over Sawyer Box because Sawyer Bin had the same state space dimension as the pre-training task). We find that this process leads to better sample efficiency (compare Fig. 18 and Fig. 12 top left), though the difference is not drastic. Overall, this experiment provides some preliminary indication that single-goal CRL representations are transferrable between tasks.

---

> > ### Comment · Reviewer_gayD · 2024-11-26
> > **Final Response**
> >
> > Thank you for your rebuttal and my sincerest apologies for the delay in response.
> >
> > "we have added new analysis on policy rollouts (see new Fig. 15)"
> >
> > Thank you for this, I have looked at this figure and while I think it is helpful it doesn't provide definitive evidence of distinct skill learning in my opinion. I think this suggests that the current approach has some promising signs of meaningful exploratory/skill learning behaviours but more work is required to develop the theory and practical use of this approach. I think the analysis of approaches that implicitly develop a curriculum remains very valuable so I'm excited to see further development on the given analysis.
> >
> > "We have also highlighted unique skill learning in a new, highly-difficult environment (see new Fig. 14) and added additional analysis on representation probing (see new Fig. 17)"
> >
> > I appreciate this thank you for adding this task environment. This addition was useful to me.
> >
> > "Do the new analysis and experiments address the reviewer's concerns about the paper? We look forward to continuing the discussion!"
> >
> > Partially I think these changes improve upon the initial submission. I do still have some reservations but my outlook on this work is more positive in that the potential ambiguities in the efficacy of the current work are outweighed by the potential novelty and usefulness of progress in this direction.
> >
> > "For all 15 random seeds that we have visualized, we always see a similar progression of skill learning. The time at which each skill is learned differs across seeds. We did not observe any seeds that behaved randomly all throughout training. Additionally, some seeds converge to different ways of solving the task (see Fig. 7). We have uploaded training videos for each environment to the anonymous code repository. These training videos correspond to one seed for each of the three sawyer environments, but we are happy to provide training videos for more seeds if requested."
> >
> > I do still feel example with more seeds are required. The current results are interesting but require more supporting qualitative evidence.

---

> > > ### Author Response · Authors · 2024-11-26
> > > **Author Response**
> > >
> > > Thank you for your response and positive feedback about the potential novelty and usefulness of this research!
> > >
> > > > more work is required to develop the theory and practical use of this approach
> > >
> > > We are also excited to develop theory for and practical methods utilizing single-goal exploration. As we mention in section 5, we are particularly interested in how we can leverage the success of SGCRL to address exploration in other problem settings, such as settings with a reward function or settings without a single-goal. We believe that the skill learning observed in the new sawyer box hard environment provides support for this line of research.
> > >
> > > > I do still feel example with more seeds are required.
> > >
> > > We have uploaded zip files of training videos for more seeds to the anonymous [code repository](https://anonymous.4open.science/r/sgcrl-C100/README.md) under the directory /training_videos. We have also included a file of videos showcasing two divergent strategies that we observe in the Sawyer bin environment.

---

> ### Author Response · Authors · 2024-11-17
> **Author Response (part 2)**
>
> > Were there any exceptions to the skill learning trends identified in (1) (e.g. checkpoints with purely random behaviours) and if so how frequent were exceptions?
>
> For all 15  random seeds that we have visualized, we always see a similar progression of skill learning.  The time at which each skill is learned differs across seeds.  We did not observe any seeds that behaved randomly all throughout training. Additionally, some seeds converge to different ways of solving the task (see Fig. 7). We have uploaded [training videos](https://anonymous.4open.science/r/sgcrl-C100/sawyer_box.mp4) for each environment to the anonymous code repository. These training videos correspond to one seed for each of the three sawyer environments, but we are happy to provide training videos for more seeds if requested.
>
> > I also wished to ask when you mention fixed sampling intervals for checkpoints, what fixed interval was used?
>
> The checkpoints were saved at fixed time intervals (e.g. every 15 minutes), which corresponded to ~3,300 trials in the environment. We have added this detail to Appendix D. The [videos](https://anonymous.4open.science/r/sgcrl-C100/sawyer_box.mp4) we uploaded also have trial numbers to give a sense of how far along in training each skill emerges.
>
>
> **References:**
>
> [1] Wu, Y., Tucker, G., & Nachum, O. (2018). The laplacian in rl: Learning representations with efficient approximations. arXiv preprint arXiv:1810.04586.
>
> [2] Erraqabi, A., Machado, M. C., Zhao, M., Sukhbaatar, S., Lazaric, A., Ludovic, D., & Bengio, Y. (2022, August). Temporal abstractions-augmented temporally contrastive learning: An alternative to the Laplacian in RL. In Uncertainty in Artificial Intelligence (pp. 641-651). PMLR.
>
> [3] Machado, M. C., Bellemare, M. G., & Bowling, M. (2017, July). A laplacian framework for option discovery in reinforcement learning. In International Conference on Machine Learning (pp. 2295-2304). PMLR.

---

### Official Review · Reviewer_zVRJ · 2024-10-30

**Soundness:** 2
**Presentation:** 3
**Contribution:** 2
**Rating:** 6
**Confidence:** 4

**Summary:**

The authors describe a curious phenomenon observed in some simulated robot manipulation environments used for RL. When using contrastive RL, using a single hard goal (one far away from the robot corresponding to task success) led to better learning outcomes than using a human designed curriculum of easy and hard goals.

This is observed across 4 environments, a Sawyer bin placing task, box placing task, peg insertion, and a 2d maze navigating task.

The authors provide a number of ablation experiments trying to study why this phenomenon occurs.

**Strengths:**

The fact that a single hard goal is sufficient with contrastive RL is certainly surprising, and the paper is upfront in not having a good explanation why this occurs. The fact it occurs over a few environments provides some evidence it is not simply a one-off occurrence.

**Weaknesses:**

I am suspicious that the results are too good - that there was no environment where using multiple goals performed best. Broadly my concern is that maybe these environments are too easy. If the model is able to succeed at the task given just the final hard goal, perhaps it's too hard to design a good dense reward or curriculum of goals to speed up learning. This wouldn't be too surprising, it's often been remarked that good partial reward design is difficult.

I'm also not sure how "single goal" the final method is. In particular, Figure 10's caption was confusing to me. It seemed to suggest that in their method, the actor loss uses multiple goals, rather than a single goal? If so, this doesn't really seem like "1 goal is all you need". Essentially I think the authors may be overgeneralizing their conclusions.

My understanding so far is this:

* A contrastive critic is learned via contrastive RL, defining reward by $\phi(s,a)^T \psi(s_f)$, where $s_f \sim Geom(1-\gamma)$ steps in the future
* When generating data, we use a single hard goal $s^*$ and act according to $\pi(a|s,s^*)$
* When updating the actor, we sample a trajectory from the replay buffer of data generated according to $\pi(a|s,s^*)$. But, for each initial $s$, we then sample $s_f \sim Geom(1-\gamma)$ within the trajectory, and apply gradient updates as if we collected data according to $\pi(a|s,s_f)$, even though we actually collected data according to $\pi(a|s,s^*)$.

In which case, at most you could say 1 goal is the only requirement for data collection, but the policy still needs to be trained on every intermediate goal observed to achieve good performance.

**Questions:**

Do the authors have any statistics on eval performance over the same distribution of goals used to generate the replay buffer? One natural argument is that, in so far as each goal can be viewed as a separate task, the model will be best at the goal distribution that appears in the replay buffer. I believe this is distinct from the overfitting experiments, because in those experiments the final evaluation number is still only the final hard eval goal.

---

> ### Author Response · Authors · 2024-11-17
> **Author Response**
>
> Dear Reviewer,
>
> Thank you for the time and effort put into the review. It seems like the main concern is about the difficulty of these tasks. We would like to emphasize that state-of-the-art prior methods not only fail to achieve good performance on these tasks, but often fail to ever reach the goal during training, highlighting the high difficulty of these tasks (see "SAC+HER" and "RIS" in Fig. 9). Second, we would like to clarify that we use "single goal" to mean that the only human supervision to the algorithm is a one goal observation; the reviewer is correct that the algorithm internally uses automatically-sampled goals for training the actor. **Together with the new analysis (see new Fig. 16) and discussion below, does this fully address the reviewer's concerns about the paper?**
>
> > Do the authors have any statistics on eval performance over the same distribution of goals used to generate the replay buffer?
>
> We have run a new experiment to study this question. For single-goal CRL, we generate the replay buffer by commanding the agent with only the single-hard goal during data-collection. For multi-goal CRL, we command the agent with goals along the path to the single-goal state. We have added evaluations for this distribution of goals in Appendix section E (see new Fig. 16), and we find that performance of single-goal CRL on this distribution of multiple goals remains high for the sawyer box and sawyer peg environments. This additional evaluation suggests that single-goal CRL retains its ability to accomplish multiple tasks without being explicitly guided to learn those tasks.
>
> > maybe these environments are too easy
>
> The robotic locomotion environments are very difficult for the strong baselines we compared to: SAC and RIS (state-of-the-art at the time of publication). Given the task difficulty, we were also surprised that multi-goal CRL could achieve higher performance with fewer assumptions.  We have also run a new experiment to test the limits of the single-goal CRL method. We created a new environment, which we call sawyer box hard, in which the agent must place a block in the box and also place the lid on top of the box (see new Fig. 14). This environment is especially difficult because the required behavior is non-linear and requires multi-step planning. While single-goal CRL was not able to consistently solve this environment (success rate < 0.05), we observed unexpected emergent skills during training which were not observed in the simpler environments. We showcase a few of these emergent skills in a video uploaded to the [anonymous code repository](https://anonymous.4open.science/r/sgcrl-C100/sawyer_box_hard.mp4), and we have added a section about this environment in Appendix C. These results further supports our main contribution that single-goal CRL drives exploration and skill learning while lifting the burden of sub-goal/reward design.
>
> > Not sure how "single goal" the final method is… Figure 10's caption was confusing because the actor loss uses multiple goals, rather than a single goal.
>
> Thanks for the suggestions for improving the presentation, which we have incorporated by revising section 3.2 (orange text). We think that this new presentation does a more effective job of clarifying that the single-goal is used for data-collection and evaluation, but multiple goals sampled from future states in the replay buffer and used for actor training. We have also updated Figure 2 to reflect this clarification. We found that using the single-goal for actor training decreased performance (see Fig. 10), indicating that multi-task learning during actor training is a key component of the method. Using the single-goal for data-collection, however, leads to exploratory behavior without requiring any rewards or curriculum of subgoals.

---

> > ### Author Response · Authors · 2024-11-21
> > **Response feedback**
> >
> > Dear Reviewer,
> >
> > We have incorporated the review feedback by running new experiments and revising the paper. We'd really appreciate if you could confirm whether these changes address the concerns about the paper. If we have misunderstood any of the concerns, we'd like to learn so that we can further revise the paper or run additional experiments.
> >
> > Best,
> > The Authors

---

> > > ### Comment · Reviewer_zVRJ · 2024-11-26
> > > **Response**
> > >
> > > I thank the authors for the updates to the text, I think this does make it more clear.
> > >
> > > I find Appendix E results interesting - I feel there is a pretty significant drop in performance when using a single hard goal and evaluating on a range of goals, and I feel this is evidence in favor of CRL-SingleHardGoal somewhat overfitting to the eval regime of using a single hard goal. In particular I think this somewhat contradicts the Figure 10 ablation figure, where a multi-goal data collection setup does worse at the single hard goal. But as seen in Figure 16, that's because the performance on the range-of-goals is higher.
> > >
> > > I plan to keep my rating the same - the fact that every non-contrastive RL approach fails to solve the problem does suggest the problem is nontrivial.

---

> > > > ### Author Response · Authors · 2024-11-28
> > > > **Author Response**
> > > >
> > > > Thank you for the response! As noted by the reviewer, the single-goal CRL performance does drop significantly when evaluated on a range of goals in the Sawyer Bin environment (see Fig. 16). This drop could indicate some overfitting to the single hard goal in this environment. However, we would like to highlight that single-goal CRL performs comparably to multi-goal CRL for both the Sawyer Box and Sawyer Peg environments when evaluated on a range of goals. Additionally, the ablation study (see Fig. 10) shows that using the single goal for policy training degrades performance, indicating that multi-task learning is essential to the effectiveness of the single goal method. Taken together, we believe that these ablation experiments help provide intuition into our main finding, which is that simply commanding a single difficult goal works at all (and works better than prior methods), even in settings where random exploration will never reach that goal.
> > > >
> > > > If the reviewer has feedback on presenting the results in Fig 16 (e.g., express that single-goal CRL could be overfitting to the single-hard goal while maintaining the importance of multi-task learning to the method), please let us know!

---

### Official Review · Reviewer_Gqvh · 2024-11-02

**Soundness:** 2
**Presentation:** 3
**Contribution:** 2
**Rating:** 6
**Confidence:** 4

**Summary:**

This work aims to solve goal-conditioned RL (GCRL) tasks where only one goal is desired without access to rewards (dense or sparse), curriculums, or demonstrations. It proposes modifying standard Contrastive RL (CRL) by using only a single goal in rollouts while still using multiple goals while training. The paper runs experiments to show that the specific combination of single-goal rollouts, multi-goal training, and contrastive representations are needed to achieve good results. The paper also includes some additional analysis to examine single-goal CRL under impossible goals and to explore the representations learned by CRL.

**Strengths:**

Under the conditions set by the work, i.e. learning single difficult goals without rewards, demonstrations, curriculums, etc, the single-goal CRL clearly outperforms other methods. Furthermore, this method reduces assumptions compared to standard RL. Finally, an important finding of this paper is that a policy can be trained under contrastive learning to find a difficult goal without providing intermediate easier goals.

**Weaknesses:**

Firstly, it is unclear how the authors selected or defined “difficult goals”. Are the difficult goals sampled by failure cases of CRL? How many difficult goals are the results evaluated on? For example, if this method is effective on one particular goal, but not others, then it would not be effective in practice.

Secondly, the authors claim that the first ablation disproves the hypothesis that single-goal CRL “learns useful representations and an effective policy only for states along that goal path” (line 476). However, since the single-goal CRL fills the replay buffer with rollouts where the policy aims to reach goal $s^*$, and the actor is only being updated with states from the replay buffer, how does this prove that the single-goal rollout + multi-goal training policy is not overfit to/does not only contain good representations for states along the goal path?

Finally, Fig. 11 shows that a monolithic critic CRL (Range of Difficulties) can reach approximately 60% success rate on the single hard goal on Sawyer Box after 100,000 steps. However, Fig. 3 shows that standard CRL (range of difficulties) can only achieve up to approximately 40% success rate on Sawyer Box after 1,000,000 steps. It seems this might contradict the assertion that the separate representations are important? (It is possible that the methodology for selecting the “difficult goals” is impacting these results.)

**Questions:**

How are the “difficult goals” sampled, how many are sampled, etc? Why are thy chosen in this way?

Are there other experiments done (e.g. something like the perturbation test, or a test where the single-goal policy shows useful representations or reasonable success on goal states outside of those in the goal path) which would alleviate concerns that this method is simply overfitting to solve for this one goal?

---

> ### Author Response · Authors · 2024-11-17
> **Author Response**
>
> Dear Reviewer,
>
> Thank you for the detailed review and constructive feedback on the paper. It seems like the main concerns relate to the conclusions from our experimental findings. We believe that the key experimental conclusions are correct; we have added a new experiment (see new Fig. 16) to verify these results, and have also revised the paper (see Sec 3.2, Sec 4.3). **Together with the responses below, does this fully address the reviewer's concerns?** We look forward to continuing the discussion!
>
> > Are there other experiments done which would alleviate concerns that this method is simply overfitting to solve for this one goal?
>
> To study this question, we did additional analysis where we evaluated the policy trained by our method on a range of goals. In particular, we evaluated our method on reaching goals sampled uniformly along the path from the starting point to the single, hard goal state. We find that single-goal CRL retains the ability to accomplish multiple tasks without being explicitly guided to learn those tasks, especially in the sawyer box and sawyer peg environments; the previous multi-goal CRL method required these subgoals to be defined explicitly. We have added this new analysis in appendix section E (see new Fig. 16). This result, in combination with the ablation experiment (Fig. 10), suggest that single-goal CRL is not overfitting its policy parameters on a single task. Instead it performs multi-task learning, which surprisingly improves performance on the single task as well compared to prior methods.
>
> Thank you also for pointing out the discrepancy in our wording. We have updated section 4.3 (orange text) to clarify that single-goal CRL “does not just overfit the policy parameters to reach the single goal, but also learns useful representations for states along the path from the starting point to the goal.” While we don't expect single-goal CRL to succeed consistently on out-of-distribution (OOD) goals (the perturbation experiment shows that encountering OOD states does degrade performance), we find that it can reach in-distribution goals that it has seen during training.
>
>
> > Unclear how the authors selected or defined “difficult goals”
>
> We have revised section 3.2 (orange text) to clarify how we select the one goal per task. Our environments all have natural semantically meaningful goal states — closing a box, inserting a peg into a narrow hole, or reaching the end of a maze. For each task, we defined the difficult goal as the respective environment coordinates for those end states. Previous methods reach this difficult goal with the help of rewards, demonstrations, or subgoals, but our method does not require those assumptions.
>
>
> > Figure 11 (right) might contradict the assertion that the separate representations are important?
>
> We have revised the wording of Sec 4.3 to clarify that this experiment shows that representations _could be_ important for driving exploration, specifically for the single-goal CRL method; this claim does not necessarily apply to the multi-goal CRL. Note that this clarification does not affect the main claim of our paper: that using a single goal for exploration is surprisingly effective and can result in the emergence of skills. We also find the the difference between monolithic critic CRL (Range of Difficulties) in Fig 11 (right) and CRL (Range of Difficulties) in Fig 3 (top right) is not statistically significant (p = 0.096).

---

> > ### Author Response · Authors · 2024-11-21
> > **Response feedback**
> >
> > Dear Reviewer,
> >
> > We have incorporated the review feedback by running new experiments and revising the paper. We'd really appreciate if you could confirm whether these changes address the concerns about the paper. If we have misunderstood any of the concerns, we'd like to learn so that we can further revise the paper or run additional experiments.
> >
> > Best,
> > The Authors

---

> > > ### Comment · Reviewer_Gqvh · 2024-11-26
> > >
> > > Thank you for your response. While 2/3 of my concerns have been addressed, my remaining concern is in how the difficult goals are sampled.
> > >
> > > In each of the proposed environments, there are multiple possible difficult terminal states. Deciding which of these difficult terminal states are used, and how many are used, will naturally affect the rest of the evaluation.
> > >
> > > For example, standard CRL does not achieve 100% success on these environments, hence there are naturally some difficult terminal states it will not be able to reach. If the difficult terminal state for training/evaluation is selected from these failure cases, then it is unsurprising that a method like single-goal CRL would outperform standard CRL in this case. However, this does not necessarily indicate single-goal CRL is the better method for achieving a single goal in general.
> > >
> > > As noted by reviewer zVRJ, it is odd that multi-goal CRL never achieves better (or even similar) performance to single-goal CRL. I am concerned this is a result of the evaluation methodology.
> > >
> > > If the authors train and evaluate on multiple possible difficult terminal states (i.e. one single-goal CRL policy per difficult terminal state), then how are these difficult terminal states sampled?
> > >
> > > If only one difficult terminal state is selected per environment, why is only one selected, and what reasoning/methodology brought authors to selected this single difficult terminal state per environment?

---

> > > > ### Author Response · Authors · 2024-11-28
> > > > **Author Response**
> > > >
> > > > Thank you for your response and further questions! We aim to clarify our motivation for how we choose the single, difficult goal below. We have also run a new experiment to study how the single-goal and multi-goal CRL methods perform when we use a single, easy goal rather than a single, difficult goal. **Does the following discussion, along with the new experiment, fully address the reviewer’s final concern?**
> > > >
> > > > > If only one difficult terminal state is selected per environment, why is only one selected, and what reasoning/methodology brought authors to selected this single difficult terminal state per environment?
> > > >
> > > > > If the difficult terminal state for training/evaluation is selected from these failure cases, then it is unsurprising that a method like single-goal CRL would outperform standard CRL in this case.
> > > >
> > > > For the environments studied here, we choose the single hard goal to be the goal state used in prior work [1] (e.g., for Sawyer Bin see [Line 55 of the Metaworld codebase](https://github.com/Farama-Foundation/Metaworld/blob/master/metaworld/envs/mujoco/sawyer_xyz/v2/sawyer_bin_picking_v2.py#L55)); this is the state corresponding to successful task completion in the Metaworld benchmark. For example, in the sawyer bin environment, the single goal state is the state in which the block is located in the middle of the blue bin. Given that the state space is continuous, we measure success the same way as [1], where an episode is successful if the object is within a given distance metric (e.g. [5 cm](https://github.com/Farama-Foundation/Metaworld/blob/master/metaworld/envs/mujoco/sawyer_xyz/v2/sawyer_bin_picking_v2.py#L99)) from the single, goal state.
> > > >
> > > > In this sense, we don’t sample from a distribution of difficult terminal states because we use only one state (the state corresponding to success) for both data collection and evaluation. In contrast, multi-goal CRL [2] is provided with subgoals sampled uniformly along the path from the starting state up to (and including) the single goal state.
> > > >
> > > > We have revised section 3.2 (orange text) to improve our presentation on the motivation behind our choice of the single-goal.
> > > >
> > > > > As noted by reviewer zVRJ, it is odd that multi-goal CRL never achieves better (or even similar) performance to single-goal CRL.
> > > >
> > > > To study this further, we ran a new experiment (see new Fig. 19)  in the point spiral environment but instead of setting the goal to be the end of the maze, we chose the goal to be a corner close to the starting point. For this single, easy goal, we observe very similar performance between multi-goal CRL and both are able to completely solve the environment (100% success) within 500,000 actor steps. This new experiment suggests that the performance difference that we observe between single-goal CRL and the other methods is due to the difficulty of the task and helps validate the correctness of our multi-goal CRL implementation.
> > > >
> > > > **References**
> > > >
> > > > [1] Yu, T., Quillen, D., He, Z., Julian, R., Hausman, K., Finn, C., & Levine, S. (2020, May). Meta-world: A benchmark and evaluation for multi-task and meta reinforcement learning. In Conference on robot learning (pp. 1094-1100). PMLR.
> > > >
> > > > [2] Eysenbach, B., Zhang, T., Levine, S., & Salakhutdinov, R. R. (2022). Contrastive learning as goal-conditioned reinforcement learning. Advances in Neural Information Processing Systems, 35, 35603-35620.

---

> > > > > ### Author Response · Authors · 2024-12-01
> > > > >
> > > > > Dear Reviewer,
> > > > >
> > > > > Do the revisions and new analysis fully address the concerns about the paper?  We believe that these changes clarify our choice of goal selection and help validate our results.
> > > > >
> > > > > Kind regards,
> > > > >
> > > > > The Authors

---

> ### Comment · Reviewer_Gqvh · 2024-12-02
>
> I appreciate the care and time from the authors' side to address my concerns, as well as the additional clarity on goal selection. However, my concerns regarding sampling the single hard goal state remain.
>
> It is worth noting that MetaWorld provides a distribution of goal states which can be sampled at the beginning of each episode (e.g. see MT1 [1]), not just a single goal state. The original CRL implementation (which the provided anonymous code seems to build on) seems to implement this as well via `self._freeze_rand_vec=False` [2].
>
> From the authors' response, it seems this paper uses the environment's "default" goal state for their single goal, rather than sampling from a distribution of possible goal states (i.e. train different single-goal CRL policies on different — but still just one — end goal state).
>
> This paper heavily relies on the assumption that single-goal CRL is better than alternatives for reaching a "single difficult goal." However, in such environments, the default goal state is often *not* the most difficult goal state from the provided goal state distribution (or even a generally difficult goal state compared to the rest of the goal state distribution).
>
> So, as I mentioned previously, since each task evaluates on only the default environment goal state, it is unclear if single-goal CRL will outperform standard CRL when selecting *any* possible goal state from the environment, including goal states which might be more difficult than the environment's default goal state.
>
> Finally, in the original CRL paper, standard CRL evaluated on all goal states in the Sawyer Bin environment achieves ~50% success rate after 3e6 steps [3]. Meanwhile, in this work, standard CRL evaluated on the Sawyer Bin default goal state achieves <20% success rate on all seeds after 1.5e7 steps (5x the number of steps). This indicates to me there are certain goal states — which may be more or less difficult than the default goal state — where standard CRL would perform similarly (or potentially better) compared to single-goal CRL.
>
> Although the authors provide an experiment in Fig. 19 which shows similar success rate on an 'easy' goal, this does not assuage my concerns that we do not know the general performance of single-goal CRL on the environment's provided goal state distribution, including potentially more difficult goal states.
>
> ---
>
> In summary, I believe the authors' choice to use the default environment goal state as the goal state used for training/evaluation to be flawed because 1) this is not necessarily the most difficult goal state (or even a generally difficult goal state compared to the rest of the goal state distribution) for each MetaWorld environment, and 2) it is difficult to know how single-goal CRL would perform compared to standard CRL when trained on any particular goal state sampled from the environments' provided goal state distribution (including more difficult goals).
>
> While these experiments on the default environment goal state show promise compared to other RL methods, I believe using only the default goal state for training/eval rather than training/evaluating single-goal CRL policies on multiple goal states from the environment's provided distribution (including ones more or less difficult than the default) is a fundamental flaw of this paper. Since this paper is mostly an empirical analysis, I find this flaw in experimental methodology quite important. Hence, I am choosing to decrease my score to 3, reject.
>
> [1] https://meta-world.github.io
>
> [2] https://github.com/rafapi/contrastive_rl/blob/main/env_utils.py
>
> [3] Eysenbach, Benjamin et al,. "Contrastive Learning as Goal-Conditioned Reinforcement Learning." NeurIPS 2022.

---

> > ### Author Response · Authors · 2024-12-03
> > **Author Response**
> >
> > Dear Reviewer,
> >
> > Thanks for being so involved in this review process. We've spent a lot of time trying to figure out where any potential confusion might be coming from. We think that it might come from the title of the paper. We titled the paper “A Single Goal is All You Need” to indicate that we only need a single goal to reach one important state. We do not intend to imply that training using a single goal teaches the agent to reach any state; we agree this is not true. As noted by the reviewer, we evaluate our method on one hard goal, which is not necessarily the most difficult goal. We will revise the paper to clarify this. We think that the main result of the paper is surprising because, before writing this paper, we thought that learning to reach a distant goal (a very sparse reward problem) would _require_ training training on nearby goals as well so that the agent could experience some successes (non-zero rewards). Even in cases where our method fails to reliably learn how to reach the one goal used for training, we observe non-trivial exploration behavior and what appear to be skills (see new Fig. 14 in Appendix C.2 and [videos](https://anonymous.4open.science/r/sgcrl-C100/sawyer_box_hard.mp4)).
> >
> > While below we dive into specific details, we hope that this discussion helps recontextualize the work.
> >
> > > MetaWorld provides a distribution of goal states which can be sampled at the beginning of each episode (e.g. see MT1 [1]).
> >
> > Yes, you're right that the metaworld environments typically randomize the goal positions. However, when we dig into the metaworld codebase, which we based our experiments off of, we see that the range is a 1 mm cube (e.g. see [line 40 of the Sawyer Bin environment](https://github.com/Farama-Foundation/Metaworld/blob/cca35cff0ec62f1a18b11440de6b09e2d10a1380/metaworld/envs/mujoco/sawyer_xyz/v2/sawyer_bin_picking_v2.py#L40)) around the single goal state that we use. To put this in context, the goals for standard CRL are sampled on the order of 10 cm. Thus, we believe it's fair to say that the metaworld environment effectively uses a single goal. We will revise the paper to clarify that it is actually a 1 mm cube.
> >
> > > The original CRL implementation (which the provided anonymous code seems to build on) seems to implement this as well via `self._freeze_rand_vec=False [2]`
> >
> > In [the link](https://github.com/rafapi/contrastive_rl/blob/main/env_utils.py#L279) provided,  `self._freeze_rand_vec` actually corresponds to randomizing the [starting position](https://github.com/Farama-Foundation/Metaworld/blob/cca35cff0ec62f1a18b11440de6b09e2d10a1380/metaworld/envs/mujoco/sawyer_xyz/sawyer_xyz_env.py#L677) of the robotic hand, not the goal position. We also set `self._freeze_rand_vec=False` (see [line 77 in the SGCRL environment file](https://anonymous.4open.science/r/sgcrl-C100/env_utils.py) to randomize the starting position. Note that the comment on [line 179](https://github.com/rafapi/contrastive_rl/blob/main/env_utils.py#L279) (`# Set False to randomize the goal position`) in the original CRL codebase is inaccurate and this comment does not appear in the SGCRL codebase.
> >
> > >  standard CRL evaluated on all goal states in the Sawyer Bin environment achieves ~50% success rate after 3e6 steps [3]. Meanwhile, in this work, standard CRL evaluated on the Sawyer Bin default goal state achieves <20% success rate  … This indicates to me there are certain goal states — which may be more or less difficult than the default goal state — where standard CRL would perform similarly (or potentially better) compared to single-goal CRL.
> >
> > Standard CRL performs data collection and evaluation using goals sampled uniformly [along the goal path](https://github.com/rafapi/contrastive_rl/blob/main/env_utils.py#L288) from the starting point to the single goal state (+- 5 mm) used in Metaworld. In terms of L2 distance from the starting point, the single goal that we use for data collection and evaluation is one of the hardest goals in this range. Therefore, the higher success rate in the standard CRL paper is likely because the evaluation goals are easier to reach than the single hard goal that we use for evaluation. However, we are not claiming that our method can reach any difficult goal in the environment. Indeed, our Sawyer Box Hard experiments (see new Fig. 14 in Appendix C.2 and [videos](https://anonymous.4open.science/r/sgcrl-C100/sawyer_box_hard.mp4)) show that SGCRL fails to reach a very hard goal but still learns complex, unexpected skills.
> >
> > Broadly, we'd like to clarify the main message of the paper: the main message is that, if there's a single goal you want to learn how to reach, one does not need a fancy exploration strategy, but simply can command that same goal over and over again. We believe this is surprising because random exploration never reaches this goal.

---

> ### Comment · Reviewer_Gqvh · 2024-12-03
>
> I thank the authors for their detailed response.
>
> > We do not intend to imply that training using a single goal teaches the agent to reach any state
>
> I would like to clarify that I understand the purpose of the method is to train one policy to reach exactly one goal state. My concerns are regarding the evaluation methodology for these policies.
>
> As noted in my original review, "if this method is effective on one particular goal, but not others, then it would not be effective in practice." Since this work only uses the default goal state from the given environments, which is usually not the most difficult success state, the experiments provided do not show whether single-goal CRL can indeed outperform standard CRL when reaching the more difficult (or easier) goal/success states provided by the MetaWorld environments.
>
> > However, when we dig into the metaworld codebase [...] self._freeze_rand_vec actually corresponds to randomizing the starting position
>
> I apologize for my confusion regarding the standard CRL implementation and the `self._freeze_rand_vec` attribute. However, it is not accurate to claim that the MetaWorld environments effectively use a single goal. As I am unsure on best practices for sharing videos on OpenReview, I have instead attached code below which is adapted from [MetaWorld's README.md to run MT1](https://github.com/Farama-Foundation/Metaworld/blob/cca35cff0ec62f1a18b11440de6b09e2d10a1380/README.md#running-ml1-or-mt1) on peg-insert-side-v2. In the generated video, one can see the box (goal) and peg (object) are both randomized far more than $1\text{mm}^3$.
>
> > Standard CRL performs data collection and evaluation using goals sampled uniformly along the goal path from the starting point to the single goal state (+- 5 mm) used in Metaworld.
>
> Are the authors certain of this? The CRL paper defines success for each environment in Appendix E.1. For Sawyer Bin, the paper states, "sawyer bin (image, state) – This task is taken from Yu et al. [139]. This task involves using a robotic arm to pick up a block from one bin and place it at a goal location in another bin. The benchmark specifies success as reaching within 5cm of the goal" [1]. The last two sentences seem to indicate that the goal locations are indeed in the other bin.
>
> However, if the authors can indeed confirm that the CRL implementation also uses goals states along the path to success to determine *success rate during evaluation*, then I am happy to reevaluate this portion of my review.
>
> [1] Eysenbach, Benjamin et al,. "Contrastive Learning as Goal-Conditioned Reinforcement Learning." NeurIPS 2022.
>
> ```
> # to run:
> # git clone https://github.com/Farama-Foundation/Metaworld.git
> # cd Metaworld
> # git checkout cca35cff0ec62f1a18b11440de6b09e2d10a1380
> # pip install -e .
> # python [file-name-here]
>
> import metaworld
> import random
> import imageio
>
> print(metaworld.MT1.ENV_NAMES)  # Check out the available environments
>
> env_name = "peg-insert-side-v2"
>
> mt1 = metaworld.MT1(env_name)  # Construct the benchmark, sampling tasks
>
> env = mt1.train_classes[env_name](
>     render_mode="rgb_array"
> )  # Create an environment
>
> # Record initial renders for 10 episodes
> all_renders = []
> for episode in range(10):
>     task = random.choice(mt1.train_tasks)
>     env.set_task(task)  # Set task
>
>     # Reset environment and record initial render
>     obs, _ = env.reset()  # Reset environment
>     all_renders.append(env.render())
>
>     # Record renders for 10 steps
>     for step in range(10):
>         a = env.action_space.sample()  # Sample an action
>         obs, reward, terminated, truncated, info = env.step(a)
>         all_renders.append(env.render())
>
> # Save all renders as one video
> imageio.mimsave("all_episodes.mp4", all_renders, fps=20)
>
> print(f"Saved {len(all_renders)} frames to video")
> ```

---

> > ### Comment · Reviewer_Gqvh · 2024-12-03
> >
> > As an update to the above, I have run the original Contrastive RL code, and indeed it seems the original method uses an altered environment where the success rate also uses intermediate goals (unlike the original MetaWorld environments). I thank the authors for making this clarification, as I had not noticed it from the original literature.
> >
> > As noted above, I will be re-evaluating this section of my review. In particular, the results from Fig. 3 in this manuscript are no longer concerning compared to the results from the original CRL paper.
> >
> > Furthermore, in my view this realization makes the results of this work more compelling: while prior contrastive RL is able to learn without rewards, it is primarily evaluated on a different (easier) success rate metric. Meanwhile, single-goal CRL is able to remove rewards *and* subgoals, all while being able to reach at least one "true" success state from the original benchmark.
> >
> > While I maintain that the evaluation methodology would benefit from training and evaluating single-goal policies on different target goals from the original MetaWorld benchmark, since prior work evaluates on an altered version of the MetaWorld benchmark as well, the single-goal CRL work restricting the selected target goal to only the default state seems more permissible.
> >
> > Hence, I would like to increase my score to a 6. I thank the authors again for their continued engagement and clarifications.

---

> > > ### Author Response · Authors · 2024-12-03
> > > **Author Response**
> > >
> > > Dear Reviewer,
> > >
> > > We really appreciate all the time and care put into this review and are grateful for the score increase. As noted by the reviewer, the CRL implementation does use goals states along the path to success to determine success rate during evaluation. We will revise section 4 to highlight this difference in evaluation methodology. Below we aim to clear up the final concern about the randomization of goals in the Metaworld environment.
> > >
> > > > it is not accurate to claim that the MetaWorld environments effectively use a single goal [...] In the generated video, one can see the box (goal) and peg (object) are both randomized far more than 1 mm^3.
> > >
> > > Thank you for pointing this out and sending code for generating videos! Yes, we misrepresented the goal distribution of the Metaworld environment by focusing on the Sawyer Bin example. In the Sawyer Bin environment, goals are randomized only [1 mm^3](https://github.com/Farama-Foundation/Metaworld/blob/cca35cff0ec62f1a18b11440de6b09e2d10a1380/metaworld/envs/mujoco/sawyer_xyz/v2/sawyer_bin_picking_v2.py#L40). The reviewer is correct that in the Sawyer Peg and Sawyer Box environment, the target object is randomized 20 cm^3 (see goal bounds for [Box](https://github.com/Farama-Foundation/Metaworld/blob/cca35cff0ec62f1a18b11440de6b09e2d10a1380/metaworld/envs/mujoco/sawyer_xyz/v2/sawyer_box_close_v2.py#L27) and [Peg](https://github.com/Farama-Foundation/Metaworld/blob/cca35cff0ec62f1a18b11440de6b09e2d10a1380/metaworld/envs/mujoco/sawyer_xyz/v2/sawyer_assembly_peg_v2.py#L28) environments). We do in fact use these randomized goals in our training and evaluation for SGCRL (e.g. see line 226 and 231 of the [SGCRL Sawyer Peg](https://anonymous.4open.science/r/sgcrl-C100/env_utils.py) code, see SGCRL [training videos](https://anonymous.4open.science/r/sgcrl-C100/sawyer_peg.mp4) which show randomized goal positions for each episode).
> > >
> > > > the evaluation methodology would benefit from training and evaluating single-goal policies on different target goals from the original MetaWorld benchmark
> > >
> > > We apologize for the confusion caused by our previous statement that we only use a single goal rather than a distribution of goals. We wanted to convey that in the experiments presented (Fig. 3), SGCRL uses a single goal - the goal instantiated when the Metaworld environment resets - rather than goals sampled along the path to success (as used in standard CRL). However, it is more precise to say that we use a single goal sampled from Metaworld’s default goal distribution. We will revise section 3.2 to clarify this, and hope that this clarification addresses the reviewer’s final concern.

---

### Meta-Review · Area_Chair_YC7Q · 2024-12-20

**Metareview:**

This paper presents an interesting phenomena in contrastive RL where a single hard goal lead to better performing policies than learning with curricula. Empirical results show that diverse skills emerge during learning long before the agent reaches the goal state and receives a good reward.

Reviewers agree this is an interesting observation that is potentially impactful for its simplicity and effectiveness. The authors are suggested to revise their presentation of the work (potentially the title) to avoid misunderstandings.

**Additional Comments On Reviewer Discussion:**

Reviewers had confusion about the problem setting and empirical evaluation but had them cleared during the discussion.

---

### Decision · Program_Chairs · 2025-01-22

Accept (Poster)